# Pairwise Learning with Adaptive Online Gradient Descent*

**Tao Sun**                                                    *suntao.saltfish@outlook.com*
*College of Computer*
*National University of Defense Technology*
*Changsha, Hunan, China*

**Qingsong Wang**                                                    *wang.8973@osu.edu*
*Department of Mathematics*
*Scientific Computing and Imaging (SCI) Institute*
*University of Utah*
*Salt Lake City, Utah, USA*

**Yunwen Lei**                                                    *leiyw@hku.hk*
*Department of Mathematics*
*The University of Hong Kong*
*Pokfulam, Hong Kong*

**Dongsheng Li**                                                    *dsli@nudt.edu.cn*
*College of Computer*
*National University of Defense Technology*
*Changsha, Hunan, China*

**Bao Wang**                                                    *wangbaonj@gmail.com*
*Department of Mathematics*
*Scientific Computing and Imaging (SCI) Institute*
*University of Utah*
*Salt Lake City, Utah, USA*

**Reviewed on OpenReview:** *https://openreview.net/forum?id=rq1SaHQg2k*

## Abstract

In this paper, we propose an adaptive online gradient descent method with momentum for pairwise learning, in which the step size is determined by historical information. Due to the structure of pairwise learning, the sample pairs are dependent on the parameters, causing difficulties in the convergence analysis. To this end, we develop novel techniques for the convergence analysis of the proposed algorithm. We show that the proposed algorithm can output the desired solution in strongly convex, convex, and nonconvex cases. Furthermore, we present theoretical explanations for why our proposed algorithm can accelerate previous workhorses for online pairwise learning. All assumptions used in the theoretical analysis are mild and common, making our results applicable to various pairwise learning problems. To demonstrate the efficiency of our algorithm, we compare the proposed adaptive method with the non-adaptive counterpart on the benchmark online AUC maximization problem.

## 1 Introduction

Let $\mathcal{K} \subseteq \mathbb{R}^d$ be a closed convex set (can be the full space $\mathbb{R}^d$) representing the parameter space. Given a statistical sample space $\Xi$ with probability distribution $\mathcal{P}$; let $F(\cdot; \xi, \xi') : \mathbb{R}^d \to \mathbb{R}$ be a closed function

---

*The first and second authors contributed equally to this paper. Dongsheng Li is the corresponding author.

Dongsheng Li and Tao Sun are supported in part by the National Science Foundation of China (62025208), Hunan Provincial Natural Science Foundation of China (2022JJ10065), Young Elite Scientists Sponsorship Program by CAST (2022QNRC001), and Continuous Support of PDL (WDZC20235250101).

associated with two samples $\xi, \xi' \in \Xi$. This paper considers the following pairwise learning problem

$$\min_{\boldsymbol{x} \in \mathcal{K} \subseteq \mathbb{R}^d} \left\{ f(\boldsymbol{x}) := \mathbb{E}_{(\xi, \xi') \sim \mathcal{P} \oplus \mathcal{P}} F(\boldsymbol{x}; \xi, \xi') \right\}, \tag{1}$$

where the function $F(\boldsymbol{x}; \xi, \xi')$ can be either convex or nonconvex in $\boldsymbol{x}$. The pairwise learning model (1) describes various classical machine learning tasks, including the metric learning (Weinberger & Saul, 2009; Kulis et al., 2013; Xing et al., 2002; Ying & Li, 2012), ranking (Rejchel, 2012; Agarwal & Niyogi, 2009), two-stage multiple kernel learning (Kumar et al., 2012), neural link prediction (Wang et al., 2021), the minimum error entropy principle (Hu et al., 2013), and can be adapted to the area under ROC curve (AUC) maximization (Zhao et al., 2011; Gao et al., 2013; Ying et al., 2016; Liu et al., 2018) (see Remark 2 in Section 3 for more details).

There are two major kinds of workhorses for the model (1), i.e., *offline* and *online*. The offline one is similar to the empirical risk minimization (ERM): given an i.i.d. sample set $\{\xi^1, \ldots, \xi^n\}$, we solve

$$\min_{\boldsymbol{x} \in \mathcal{K}} \frac{1}{n(n-1)} \sum_{i,j \in [n], i \neq j}^{n} F(\boldsymbol{x}; \xi^i, \xi^j),$$

where $[n] := \{1, 2, \ldots, n\}$. The major difference between the offline pairwise learning model and the ERM lies in the efficiency of the samples and whether the objective functions are independent of each other. A $n$-samples training set outputs a finite-sum minimization with $\mathcal{O}(n)$ sub-functions in ERM, while the same training set results in $\mathcal{O}(n^2)$ in the offline pairwise learning. Furthermore, the objective functions in the ERM are independent of each other, which is broken for the offline pairwise learning[1]. One possible modification to circumvent the dependent objective functions is to form the objective function with two new independent samples in each iteration. This method is suggested in (Peel et al., 2010, Section 4.2) and can be regarded as the algorithm proposed by Zhao et al. (2011) with buffer size one. However, as shown in (Zhao et al., 2011), this two-dependent-points version of online learning does not fully utilize the sampling sequence and results in inferior performance compared with algorithms that utilize some historical samples.

The online pairwise learning assumes the i.i.d. samples $(\xi^k)_{k \in [n]}$ are continuously received by the model. In the $k$th iteration, the online style algorithm proceeds to sample new data $\xi^k$ from $\mathcal{P}$ and reuses the previous samples $(\xi^{j_i})_{1 \leq i \leq s}$ with $\{j_i\}_{1 \leq i \leq s} \subseteq [k-1]$ to get the mini-batch stochastic gradient $\boldsymbol{g}^k = \frac{1}{s} \sum_{i=1}^{s} \nabla F(\boldsymbol{x}^k; \xi^k, \xi^{j_i})$ (Wang et al., 2012; Zhao et al., 2011; Ying & Zhou, 2016). Thus, the online method needs to employ an $\mathcal{O}(s)$ memory to store the previously sampled data and $\mathcal{O}(s)$ computations to calculate the gradient. Nevertheless, the mini-batch version suffers two drawbacks: 1) It has been proved that its excess generalization bound can be as large as $\mathcal{O}\left(\frac{1}{\sqrt{s}} + \frac{1}{\sqrt{n}}\right)$ (Wang et al., 2012), and we need to use a large $s$ to improve generalization of the online method. 2) A large $s$ causes tremendous computational and memory costs that are unacceptable for online settings. To this end, a simple yet efficient online gradient descent (OGD) is proposed (presented as Algorithm 1), in which the stochastic gradient $\boldsymbol{g}^k$ is set to be $\nabla F(\boldsymbol{x}^k; \xi^k, \xi^{k-1})$ (Yang et al., 2021b). An interesting finding is that the OGD can achieve $\mathcal{O}\left(\frac{1}{\sqrt{n}}\right)$ excess generalization bound. The favorable memory and computation costs make the OGD applicable to broader online settings.

In this paper, we focus on developing provably convergent online algorithms with adaptive stepsizes for pairwise learning. Considering the efficiency of the sampling method of OGD (Yang et al., 2021b), our algorithm inherits such kind of sampling. Moreover, our algorithm employs momentum.

## 1.1 The Adaptive Online Gradient Descent for Pairwise Learning

The OGD performs an SGD-style (stochastic gradient descent) iteration but with biased stochastic gradients since $\xi^{k-1}$ is related to $\boldsymbol{x}^k$. Motivated by the remarkable success of the adaptive variants of SGD for machine learning (Duchi et al., 2011; McMahan & Streeter, 2010; Tieleman & Hinton, 2012; Kingma & Ba, 2015; Reddi et al., 2018; Ward et al., 2019), we propose the adaptive variant of OGD (AOGD) for pairwise learning,

---

[1]For example, $F(\boldsymbol{x}; \xi^1, \xi^2)$ and $F(\boldsymbol{x}; \xi^1, \xi^3)$ are not independent because they have a shared data $\xi^1$.

---

**Algorithm 1** Online Gradient Descent (OGD) for Pairwise Learning (Yang et al., 2021b)

---

**Parameters**: $\eta > 0$.
**Initialization**: $\boldsymbol{x}^0 = \boldsymbol{0}$, $\xi^1 \sim \mathcal{P}$
**for** $k = 1, 2, 3, \ldots$
**step 1**: receive $\xi^k$ and calculate $\boldsymbol{g}^k = \nabla F(\boldsymbol{x}^k; \xi^k, \xi^{k-1})$
**step 2**: $\boldsymbol{x}^{k+1} = \mathbf{Proj}_{\mathcal{K}}(\boldsymbol{x}^k - \eta \boldsymbol{g}^k)$
**End for**

---

---

**Algorithm 2** Adaptive Online Gradient Descent (AOGD) for Pairwise Learning

---

**Parameters**: $\eta > 0$, $0 \le \theta < 1$.
**Initialization**: $\boldsymbol{x}^0 = \boldsymbol{m}^0 = \boldsymbol{0}$, $\xi^1 \sim \mathcal{P}$
**for** $k = 1, 2, 3, \ldots$
**step 1**: receive $\xi^k$ and calculate $\boldsymbol{g}^k = \nabla F(\boldsymbol{x}^k; \xi^k, \xi^{k-1})$
**step 2**: $\boldsymbol{m}^k = \theta \boldsymbol{m}^{k-1} + (1 - \theta) \boldsymbol{g}^k$
**step 3**: $v^k = v^{k-1} + \|\boldsymbol{g}^k\|^2$
**step 4**: $\boldsymbol{x}^{k+1} = \mathbf{Proj}_{\mathcal{K}}(\boldsymbol{x}^k - \eta \boldsymbol{m}^k / (v^k)^{\frac{1}{2}})$
**End for**

---

presented as Algorithm 2. AOGD directly uses the historical sum of the moment rather than the weighted average form used in Adam (Kingma & Ba, 2015), and AOGD can be rewritten in the weighted average form: let $\hat{v}^k := \sum_{i=1}^{k} \|\boldsymbol{g}^i\|^2 / k = v^k / k$, then steps 3 and 4 of AOGD can be reformulated as follows:

$$
\hat{v}^k = \left(1 - \frac{1}{k}\right) \hat{v}^{k-1} + \frac{1}{k} \|\boldsymbol{g}^k\|^2,
$$
$$
\boldsymbol{x}^{k+1} = \mathbf{Proj}_{\mathcal{K}}\left(\boldsymbol{x}^k - \frac{\eta}{\sqrt{k}} \frac{\boldsymbol{m}^k}{(\hat{v}^k)^{1/2}}\right).
\tag{2}
$$

In reformulation (2), weights of the moment are $\{1/k\}_{k \ge 1}$ and stepsizes are $\{\eta/\sqrt{k}\}_{k \ge 1}$, satisfying the sufficient conditions for the convergence of Adam-type algorithms (Zou et al., 2019; Chen et al., 2019).

Compared with OGD, AOGD employs the momentum and adaptive stepsize generated by the historical information. Thus unlike OGD, AOGD needs a memory of history, but without much computational overhead since AOGD only computes gradient once in each iteration. Another difference between OGD and AOGD lies in the hyper-parameter $\eta$: in OGD, $\eta$ shall be set as small as the desired error $\epsilon$; while in AOGD, $\eta$ can be set as a constant that is independent of $\epsilon$.

### 1.2 Comparison with A Closely Related Work

In (Ding et al., 2015), by adopting the AdaGrad-style adaptive gradient update, Duchi et al. (2011) have proposed an adaptive method for online AUC maximization, which is a kind of pairwise learning. Although both (Ding et al., 2015) and our paper consider the adaptive online method for pairwise learning, there are four major differences between (Ding et al., 2015) and our paper, summarized below.

1) We follow the sampling method used by the OGD algorithm in (Yang et al., 2021b).
2) We consider more general cases and provide the corresponding theoretical guarantees, including the more general model (pairwise learning rather than only AUC maximization), and convergence for more general settings, including the nonconvex case, and more general schemes e.g., the use of momentum.
3) We get rid of using the regret bound because it does not directly tell us whether the algorithm converges to the desired minimizer. Another important reason for not using the regret bound for the analysis is that the regret bound has difficulties in covering the nonconvex cases. Based on the above reasons, a non-regret analysis is necessary.
4) We develop new analysis techniques to get the non-regret convergence analysis. Notice that the regret bound is not affected by the stochasticity of the data, and thus the analysis in (Ding et al., 2015) does not need to consider how to deal with the biased stochastic gradients.

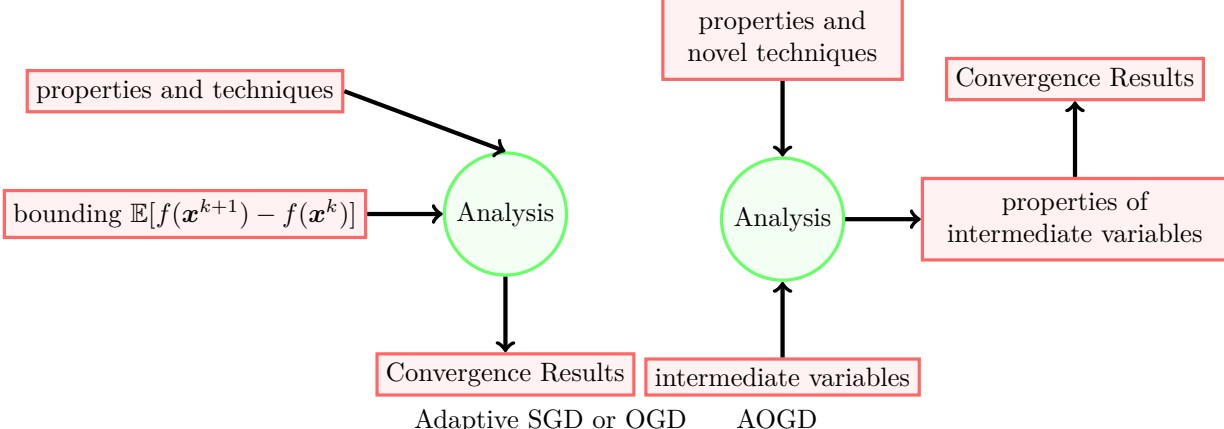

Figure 1: Contrasting the analysis of AOGD against adaptive SGD/OGD. Our analysis of AOGD studies the properties of some intermediate variables. To this end, we develop novel techniques. In the last step, we use the properties of the intermediate variables to derive the convergence guarantee for AOGD.

### 1.3 Challenges in the Analysis and Difference from Existing Analysis

The primary source of challenges in theoretical analysis comes from the fact that $\boldsymbol{x}^k$ *is dependent on* $\xi^{k-1}$, which immediately breaks the unbiased expectation of the stochastic gradient $\nabla F(\boldsymbol{x}^k; \xi^k, \xi^{k-1})$, i.e., $\mathbb{E}\big[\nabla F(\boldsymbol{x}^k; \xi^k, \xi^{k-1})\big] \neq \nabla f(\boldsymbol{x}^k)$. As such, we cannot directly follow the techniques from adaptive SGD (Duchi et al., 2011; Reddi et al., 2018; Chen et al., 2019; Ward et al., 2019; Zou et al., 2019; Li & Orabona, 2019). In paper (Yang et al., 2021b), the authors consider the following decomposition

$$\eta \nabla F(\boldsymbol{x}^k; \xi^k, \xi^{k-1}) = \eta \nabla F(\boldsymbol{x}^{k-1}; \xi^k, \xi^{k-1}) + \eta \nabla F(\boldsymbol{x}^k; \xi^k, \xi^{k-1}) - \eta \nabla F(\boldsymbol{x}^{k-1}; \xi^k, \xi^{k-1}).$$

Notice that $\boldsymbol{x}^{k-1}$ is independent of $\xi^k, \xi^{k-1}$, and $\mathbb{E}\big[\eta \nabla F(\boldsymbol{x}^{k-1}; \xi^k, \xi^{k-1})\big] = \eta \nabla f(\boldsymbol{x}^{k-1})$. The Lipschitz property of the gradient then gives us

$$\left\| \eta \nabla F(\boldsymbol{x}^k; \xi^k, \xi^{k-1}) - \eta \nabla F(\boldsymbol{x}^{k-1}; \xi^k, \xi^{k-1}) \right\| = \mathcal{O}(\eta \|\boldsymbol{x}^k - \boldsymbol{x}^{k-1}\|),$$

and the proof is similar to the "delayed" SGD.

However, our proof cannot directly follow the technique above because AOGD involves two extra recipes, i.e., momentum and adaptive stepsize. When momentum exists, we need to deal with both $\boldsymbol{g}^k$ and $\boldsymbol{m}^k$ rather than only $\boldsymbol{g}^k$, and we have to modify the techniques from (Yang et al., 2021b) to analyze the effects of momentum [2]. Another difficulty is the use of the adaptive stepsize variable $\eta/(v^k)^{\frac{1}{2}}$. Furthermore, $v^k$ is dependent of the pair $(\xi^k, \xi^{k-1})$, making the analysis more challenging.

In contrast to the existing analysis, our approach is not directly establishing the Lyapunov descent starting from bounding $\mathbb{E}[f(\boldsymbol{x}^{k+1}) - f(\boldsymbol{x}^k)]$. Instead, we recruit some intermediate variables. Taking the nonconvex case as an example, we first establish a *Lyapunov-like* descent property as follows

$$\mathbb{E}\langle -\nabla f(\boldsymbol{x}^k), \boldsymbol{m}^k/(v^k)^{\frac{1}{2}}\rangle \leq \theta \mathbb{E}\langle -\nabla f(\boldsymbol{x}^{k-1}), \boldsymbol{m}^{k-1}/(v^{k-1})^{\frac{1}{2}}\rangle + \varphi(\boldsymbol{x}^k, \boldsymbol{x}^{k-1}, \xi^k, \xi^{k-1}, \xi^{k-2}),$$

where $\varphi(\boldsymbol{x}^k, \boldsymbol{x}^{k-1}, \xi^k, \xi^{k-1}, \xi^{k-2})$ is a function of variables $\boldsymbol{x}^k$, $\boldsymbol{x}^{k-1}$, $\xi^k$, $\xi^{k-1}$, and $\xi^{k-2}$. We stress that the descent is not Lyapunov because $\theta \neq 1$ and $\varphi(\boldsymbol{x}^k, \boldsymbol{x}^{k-1}, \xi^k, \xi^{k-1}, \xi^{k-2})$ is not always negative. Then, we build the correspondence between the mathematical convergence measurement and $\mathbb{E}\big[\langle -\nabla f(\boldsymbol{x}^k), \boldsymbol{m}^k/(v^k)^{\frac{1}{2}}\rangle\big]$. A big picture of the difference in analyzing adaptive SGD/OGD and AOGD is presented in Figure 1.

### 1.4 Contributions

Our major contributions are threefold, which are summarized below.

---

[2]Indeed, in the conclusion part of (Yang et al., 2021b), the authors have listed the momentum variant as future work.

- We propose an adaptive online gradient descent algorithm for pairwise learning with a simple sampling strategy. The proposed algorithm uses adaptive stepsize and momentum, requiring only a small overhead in memory and computational costs.
- We present the convergence results for the proposed algorithm under different settings, including strongly convex, general convex, and nonconvex cases. The use of adaptive stepsize and momentum requires non-trivial techniques for the convergence analysis. We also provide theoretical explanations for why our proposed algorithm can accelerate OGD.
- We verify the efficiency of the proposed AOGD on the benchmark online AUC maximization task, showing that AOGD outperforms OGD.

## 1.5 Notation

Throughout this paper, we use boldface letters to denote vectors, e.g., $\boldsymbol{x}, \boldsymbol{y} \in \mathbb{R}^d$. The $j$th coordinate of the vector $\boldsymbol{x}$ is denoted by $\boldsymbol{x}_j$. The $L_2$ norm of the vector $\boldsymbol{x}$ is denoted by $\|\boldsymbol{x}\|$. We denote $\mathbb{E}[\cdot]$ as the expectation with respect to the underlying probability space. We denote the minimum value of the function $f$ over $\mathcal{K}$ as $\min_{\mathcal{K}} f$, and denote $\mathbf{Proj}_{\mathcal{K}}(\boldsymbol{x})$ as the projection of $\boldsymbol{x}$ onto the set $\mathcal{K}$. For two positive sequences $(a_k, b_k)_{k \geq 0}$, $a_k = \mathcal{O}(b_k)$ means that there exists $C > 0$ such that $a_k \leq Cb_k$. The notation $a_k = \Theta(b_k)$ indicates that $a_k = \mathcal{O}(b_k)$ and $b_k = \mathcal{O}(a_k)$. We use $a_k = \tilde{\mathcal{O}}(b_k)$ and $a_k = \tilde{\Theta}(b_k)$ to hide the logarithmic factor but still with the same order. We use $a_k \geq \Theta(b_k)$ to present the relation $a_k \geq Cb_k$ with $C > 0$.

## 1.6 Organization

We organize this paper as follows: In Section 2, we present assumptions and theoretical convergence results for AOGD under general convex, strongly convex, and nonconvex settings. We numerically verify the efficiency of AOGD and compare it to the benchmark OGD in Section 3. More related works are discussed in Section 4, followed by concluding remarks. The detailed proofs are provided in the supplementary materials.

# 2 Convergence Analysis

## 2.1 Assumptions

We first collect several necessary assumptions for the convergence analysis of AOGD:

- **Assumption 1**: *The function $F(\cdot; \xi, \xi')$ is differentiable, and its gradient is L-Lipschitz, i.e.,*

$$\|\nabla F(\boldsymbol{x}; \xi, \xi') - \nabla F(\boldsymbol{y}; \xi, \xi')\| \leq L\|\boldsymbol{x} - \boldsymbol{y}\|, \ \forall \boldsymbol{x}, \boldsymbol{y} \in \mathcal{K}, \xi, \xi' \in \Xi. \tag{3}$$

- **Assumption 2**: *The gradient of $F(\boldsymbol{x}; \xi, \xi')$ is uniformly bounded, i.e., $\|\nabla F(\boldsymbol{x}; \xi, \xi')\| \leq B$ for some constant $B > 0$, $\forall \boldsymbol{x} \in \mathcal{K}$, and $\xi, \xi' \in \Xi$.*

The Lipschitz smooth gradient assumption is widely used in the (non)convex optimization and pairwise learning communities. While Assumption 2 is frequently used in the adaptive SGD community, see, e.g., (Duchi et al., 2011; Reddi et al., 2018; Chen et al., 2019; Ward et al., 2019; Zou et al., 2019; Li & Orabona, 2019) [3]. Note that when $\mathcal{K}$ is bounded – Duchi et al. (2011); Reddi et al. (2018) have assumed the boundedness of the constrained set – Assumption 2 directly holds for the continuity of the gradient [4], but not vice versa. Moreover, Assumption 2 indicates the following estimate of $\nabla f(\boldsymbol{x})$:

$$\|\nabla f(\boldsymbol{x})\| = \|\mathbb{E}_{\xi, \xi' \sim \mathcal{P} \oplus \mathcal{P}} \nabla F(\boldsymbol{x}; \xi, \xi')\| \leq \mathbb{E}_{\xi, \xi' \sim \mathcal{P} \oplus \mathcal{P}} \|\nabla F(\boldsymbol{x}; \xi, \xi')\| \leq B.$$

Using mathematical induction, we can see that the momentum $\boldsymbol{m}^k$ also enjoys the uniform bound according to Assumption 2. Furthermore, we stress that we do not need to assume the variance is bounded, which is indicated by Assumption 2 since we have

$$\mathbb{E}\|\nabla F(\boldsymbol{x}; \xi, \xi') - \nabla f(\boldsymbol{x})\|^2 \leq \mathbb{E}\|\nabla F(\boldsymbol{x}; \xi, \xi')\|^2 \leq B^2.$$

---

[3] The uniform assumption presented in (Li & Orabona, 2019) enjoys another presentation, i.e., (Assumption (H2)) presented as $|f(\boldsymbol{x}) - f(\boldsymbol{y})| \leq G\|\boldsymbol{x} - \boldsymbol{y}\|$. Note that when $f$ is differentiable, it is equivalent to $\sup_{\boldsymbol{x}} \|\nabla f(\boldsymbol{x})\| \leq G$. With bounded assumption for $F(\boldsymbol{x}; \xi) - \nabla f(\boldsymbol{x})$ in (Li & Orabona, 2019), which is indeed the uniform bound assumption.

[4] Any continuous function is uniformly bounded over a closed bounded subset of $\mathbb{R}^d$.

Assumptions 1 and 2 will be used in the analysis of AOGD for all different scenarios in the subsequent analysis.

## 2.2 General Convex Cases

In this subsection, we present the convergence result of AOGD for the general convex case, i.e., $F(\boldsymbol{x}; \xi, \xi')$ is convex with respect to $\boldsymbol{x}$ and any fixed $\xi, \xi'$. Note that the convexity of $F(\boldsymbol{x}; \xi, \xi')$ indicates the convexity of $f(\boldsymbol{x})$ in (1), but not vice versa.

**Theorem 1** *(General Convexity) Let Assumption 1 hold, and $\|\boldsymbol{g}^0\|^2 \geq \delta > 0$ for some constant $\delta$, and $F(\boldsymbol{x}; \xi, \xi')$ be convex. Assume $\{\boldsymbol{x}^k\}_{k \geq 1}$ is generated by AOGD for pairwise learning, $\boldsymbol{x}^* := \arg\min_{\boldsymbol{x} \in \mathcal{K}} f(\boldsymbol{x})$, and $\mathcal{K}$ is additionally bounded, i.e., $\max_{\boldsymbol{x}, \boldsymbol{y} \in \mathcal{K}} \|\boldsymbol{x} - \boldsymbol{y}\| \leq D$. Then we have*

$$\mathbb{E}\left[ f\left( \frac{\sum_{k=1}^{K} \boldsymbol{x}^k}{K} \right) - \min_{\mathcal{K}} f \right] \leq \frac{c_1 + c_2(K)}{K}, \tag{4}$$

*where $c_1 := \frac{\mathbb{E}\sqrt{v^1}\|\boldsymbol{x}^* - \boldsymbol{x}^1\|^2}{2(1-\theta)\eta} + \frac{3B^2}{\sqrt{\delta}}$, and $c_2(K) := \left[ \frac{\eta + 2D^2}{2(1-\theta)} + (\eta\theta + 1) \right] \cdot \mathbb{E}\sqrt{v^{K+1}} + \left[ 4L + \frac{2L^2}{\sqrt{\delta}} \right] \ln \frac{\mathbb{E}\sqrt{v^{K+1}}}{\sqrt{\delta}}$.*

Assumption 2 is implicitly used in Theorem 1 because we have assumed the boundedness of the constrained set $\mathcal{K}$. The bounded constrained set is indeed stronger than the uniform bounded gradient assumption. We leave how to relax the bounded set assumption as future work. From Theorem 1, we can see that the convergence rate is dependent on $\mathbb{E}[\sqrt{v^K}]$. The boundedness of the stochastic gradients indicates that $\mathbb{E}[\sqrt{v^K}] = \mathcal{O}(\sqrt{K})$, which means the worst convergence rate of AOGD is $\widetilde{\mathcal{O}}\left( \frac{1}{\sqrt{K}} \right)$. We notice that the convergence rate $\widetilde{\mathcal{O}}\left( \frac{1}{\sqrt{K}} \right)$ coincides with the rate of OGD for pairwise learning in the general convex case (Yang et al., 2021b). However, in some cases $\mathbb{E}[\sqrt{v^K}]$ can decay faster than $\mathcal{O}(\sqrt{K})$, based on which we can establish an accelerated rate of AOGD.

**Proposition 1** *Assume the conditions of Theorem 1 hold, and assume $\mathbb{E}[\sqrt{v^K}] = \mathcal{O}(K^\alpha)$ with $0 < \alpha \leq \frac{1}{2}$. Then we have*

$$\mathbb{E}\left[ f\left( \frac{\sum_{k=1}^{K} \boldsymbol{x}^k}{K} \right) - \min_{\mathcal{K}} f \right] = \widetilde{\mathcal{O}}\left( \frac{1}{K^{1-\alpha}} \right). \tag{5}$$

If $\alpha < \frac{1}{2}$, the convergence rate of AOGD is thus better than $\widetilde{\mathcal{O}}\left( \frac{1}{\sqrt{K}} \right)$. Proposition 1 then provides a theoretical interpretation of why AOGD is possible to be faster than OGD (Yang et al., 2021b).

**Remark 1** *Note that the condition $\alpha = 1/2$ directly holds due to the boundedness of the stochastic gradients. The fast decaying condition $\alpha < 1/2$ has been commonly used as a standard explanation for why adaptive stochastic optimization algorithms often outperform non-adaptive schemes, as shown in several studies (Reddi et al., 2018; Chen et al., 2018; 2019; Liu et al., 2019). As far as our current knowledge goes, the superiority of adaptive SGD over non-adaptive approaches is not well-explained, apart from the assumption that $\alpha < 1/2$. This assumption is commonly used in previous works because many training tasks involve sparse stochastic gradients. However, it is important to note that this assumption is not a logical consequence of sparse gradients resulting in $\alpha < 1/2$. Instead, the assumption is more of a hypothetical analysis. In summary, while the assumption $\alpha < 1/2$ is widely employed in the literature, we do not have a definitive explanation for the effectiveness of adaptive SGD beyond this assumption. Further research is required to gain a comprehensive understanding of the benefits of adaptive SGD compared to non-adaptive methods.*

## 2.3 Strongly Convex Cases

In this subsection, we consider the case that the function $F(\boldsymbol{x}; \xi, \xi')$ is $\nu$-strongly convex for some constant $\nu > 0$, i.e., $F(\boldsymbol{x}; \xi, \xi') - F(\boldsymbol{y}; \xi, \xi') - \langle \nabla F(\boldsymbol{y}; \xi, \xi'), \boldsymbol{x} - \boldsymbol{y} \rangle \geq \frac{\nu}{2}\|\boldsymbol{x} - \boldsymbol{y}\|^2$. In particular, if $\nu = 0$, $F(\boldsymbol{x}; \xi, \xi')$ then reduces to general convex.

**Stepsize rule for the strongly convex case.** Before developing a convergence guarantee of AOGD for strongly convex cases, we need to select the appropriate stepsize rule for AOGD. We first need to explain the stepsize rule of AOGD for the general convex case, i.e., $\eta/\sqrt{v^k}$ for some constant $\eta > 0$, is inappropriate for solving strongly convex problems. The boundedness of gradient directly gives us $\frac{1}{\sqrt{v^k}} \geq \Theta\left(\frac{1}{\sqrt{k}}\right)$. However, such a stepsize rule causes the accumulation of stochastic noise; it will not improve the convergence rate of AOGD under strong convexity compared to the rate of AOGD under general convexity. A proper stepsize choice for the strongly convex case is $\Theta\left(\frac{1}{k}\right)$ (Bach & Moulines, 2013). Indeed, in papers (Duchi et al., 2011; Sun et al., 2020), the authors use the $\frac{1}{\sqrt{k}\sqrt{v^k}}$ stepsize rule for AdaGrad when the problem is strongly convex, and we follow this stepsize rule for the strongly convex online pairwise learning. Note that $\frac{1}{\sqrt{k}\sqrt{v^k}} \geq \Theta\left(\frac{1}{k}\right)$, which coincides with the stepsize used for SGD when the underlying problem is strongly convex. In summary, in the strongly convex case, we replace **step 4** of Algorithm 2 with **step 4'**, which is given below

$$\textbf{step 4'}: \ \boldsymbol{x}^{k+1} = \textbf{Proj}_{\mathcal{K}}\left(\boldsymbol{x}^k - \frac{\eta}{\sqrt{k}}\boldsymbol{m}^k/\sqrt{v^k}\right). \tag{6}$$

Next, we present the convergence rate of AOGD for pairwise learning with the strong convexity assumption.

**Theorem 2** *Let Assumption 1 hold, and $\|\boldsymbol{g}^0\|^2 \geq \delta > 0$, and $F(\boldsymbol{x};\xi,\xi')$ be strongly convex. Assume $\{\boldsymbol{x}^k\}_{k\geq 1}$ is generated by the AOGD for pairwise learning with $\theta = 0$ using stepsize rule **step 4'**, and $\boldsymbol{x}^* = \arg\min_{\boldsymbol{x}\in\mathcal{K}} f(\boldsymbol{x})$. By setting $\eta = \frac{B}{2\nu}$, then we have*

$$\mathbb{E}\left[\|\boldsymbol{x}^K - \boldsymbol{x}^*\|^2\right] = \mathcal{O}\left(\frac{\ln K}{K}\right). \tag{7}$$

In the strongly convex case, Assumption 2 is equivalent to the boundedness assumption of the constrained set $\mathcal{K}$. This is because the function $f(\boldsymbol{x})$ is also $\nu$-strongly convex [5], yielding $\|\nabla f(\boldsymbol{x})\| = \|\nabla f(\boldsymbol{x}) - \nabla f(\boldsymbol{x}^\dagger)\| \geq \nu\|\boldsymbol{x} - \boldsymbol{x}^\dagger\|$ with $\boldsymbol{x}^\dagger$ being the global minimizer of $f$. When Assumption 2 holds, in Subsection 2.1 we have shown that $\|\nabla f(\boldsymbol{x})\| \leq B$ over $\mathcal{K}$. Thus, we get $\|\boldsymbol{x}\| \leq \|\boldsymbol{x} - \boldsymbol{x}^\dagger\| + \|\boldsymbol{x}^\dagger\| \leq \frac{B}{\nu} + \|\boldsymbol{x}^\dagger\|$ when $\boldsymbol{x} \in \mathcal{K}$. Notice that $\mathbf{x}^\dagger$ is fixed, and the set $\mathcal{K}$ is then uniformly bounded.

Theorem 2 above shows that AOGD can achieve a faster convergence rate under the strong convexity assumption than that for general convex cases. Theorem 2 also shows that in the strongly convex case, AOGD achieves an almost optimal convergence rate of SGD under strong convexity, specifically $\widetilde{\mathcal{O}}\left(\frac{1}{K}\right)$ (the optimal convergence rate is $\mathcal{O}\left(\frac{1}{K}\right)$ according to (Rakhlin et al., 2012)). The result in Theorem 2 does not generalize to the general convex case since we set $\eta = \frac{B}{2\nu}$, which is infinity when $\nu = 0$. For technical reasons, we set $\theta = 0$ in Theorem 2, i.e., we only consider the momentum-free case. We will consider how to build the convergence rate $\widetilde{\mathcal{O}}\left(\frac{1}{K}\right)$ for AOGD with momentum in our future work.

## 2.4 Nonconvex Cases

In this part, we consider the case when $F(\boldsymbol{x};\xi,\xi')$ is nonconvex. The assumptions for the nonconvex case are much milder than the convex and strongly convex cases. We can even get rid of using the projection operator $\textbf{Proj}_{\mathcal{K}}(\cdot)$ for AOGD. The convergence result of AOGD for the nonconvex case is presented as follows.

**Theorem 3** *Let Assumptions 1, 2 hold, and let $\{\boldsymbol{x}^k\}_{k\geq 1}$ be generated by AOGD, and $\|\boldsymbol{g}^0\|^2 \geq \delta$ for some constant $\delta > 0$. Suppose $\sqrt{v^k} \leq C \cdot k^\alpha$ for two constants $C > 0$ and $0 < \alpha \leq \frac{1}{2}$, and $\mathcal{K}$ is the full space. Then, we have*

$$\min_{1\leq k\leq K}\left\{\mathbb{E}\|\nabla f(\boldsymbol{x}^k)\|^2\right\} \leq \frac{c_3 + c_4(K)}{K^{1-\alpha}}, \tag{8}$$

---

[5]Taking expectation of both sides of the inequality $F(\boldsymbol{x};\xi,\xi') - F(\boldsymbol{y};\xi,\xi') - \langle \nabla F(\boldsymbol{y};\xi,\xi'), \boldsymbol{x} - \boldsymbol{y}\rangle \geq \frac{\nu}{2}\|\boldsymbol{x} - \boldsymbol{y}\|^2$ gives us the strong convexity of $f$.

*where* $c_3 := \frac{2(7-6\theta)B^2C}{(1-\theta)\sqrt{\delta}} + \frac{4Cf(\boldsymbol{x}^1)}{\eta}$, *and* $c_4(K) := \left(\frac{(1+\theta)\eta}{1-\theta} + 2L^2/\sqrt{\delta} + L^2/\delta + 5/2\right)\ln\left(\frac{CK}{\delta}\right)$.

From Theorem 3, we can see that the convergence rate of AOGD is $\widetilde{\mathcal{O}}\left(\frac{1}{K^{1-\alpha}}\right)$, and $\alpha = \frac{1}{2}$ is the worst case due to the boundedness of the stochastic gradient. When $\alpha < \frac{1}{2}$, we get a faster convergence of AOGD compared with OGD or SGD in the general nonconvex case.

The conditions of Theorem 3 can also be satisfied by the general convex case without projection. Therefore, (8) also holds when $F(\boldsymbol{x};\xi,\xi')$ is convex. However, (8) is weaker than (4) since the convergence rate in (8) is not established with respect to the function values.

## 3  Numerical Results

Table 1: Statistics of the dataset used for contrasting the performance of AOGD and OGD, where $n$ is the number of samples in each dataset, and $d$ is the number of features of each instance in a given dataset. All datasets come from the LIBSVM website (Chang & Lin, 2011), and they are used in (Yang et al., 2021b).

|   | diabtes | german | ijcnn1 | letter | mnist | usps |
|---|---------|--------|--------|--------|-------|------|
| n | 768 | 1,000 | 49,990 | 15,000 | 60,000 | 7,291 |
| d | 8 | 24 | 22 | 161 | 780 | 256 |

In this section, we numerically validate our theoretical findings for both convex and nonconvex cases. To this end, we compare our proposed AOGD against the baseline OGD proposed in (Yang et al., 2021b) [6] for pairwise learning in terms of generalization and rate of convergence with respect to the number of iteration. We also include the results of AdaOAM in (Ding et al., 2015), an adaptive online algorithm for AUC maximization. However, we note that the algorithm in (Ding et al., 2015) is not designed for general pairwise learning. Following (Yang et al., 2021b), we consider six benchmark datasets, summarized in Table 1. Also, following the data split strategy used in (Yang et al., 2021b), for the dataset with multiple classes, we convert the first half of classes to the positive class and the second half of classes to the negative class.

For each dataset, we use 80% of the data for training and the remaining 20% for testing. All the reported results are based on 25 runs with random shuffling. The generalization performance is reported using the average AUC score and standard deviation on the test data. To determine proper hyperparameters for OGD, AOGD, and AdaOAM, we conduct 5-fold cross-validation on the training sets: 1) for OGD, we select stepsizes $\eta_t = \eta \in 10^{[-5:5]}$ [7] and the parameter space $\mathcal{K}$ is set to be the $L^2$-ball centered at the origin with radius $R \in 10^{[-3,3]}$; 2) for AOGD, we let $\theta = 0.9$ and we select stepsizes $\eta_t = \eta \in 10^{[-5:5]}$ and the parameter space $\mathcal{K}$ is also the $L^2$-ball centered at the origin with radius $R \in 10^{[-3,3]}$; 3) for AdaOAM, we select stepsizes $\eta_t = \eta \in 10^{[-5:5]}$ and the parameter space $\mathcal{K}$ is set to be the $L^2$-ball centered at the origin with radius $R \in 10^{[-3,3]}$.

**Convex case:**  We run experiments on AUC maximization using the following convex loss function

$$f(\boldsymbol{w};(\boldsymbol{x},y),(\boldsymbol{x}',y')) = \ell(\boldsymbol{w}^\top(\boldsymbol{x}-\boldsymbol{x}'))\mathbb{I}_{[y=1\wedge y'=-1]} + \ell(\boldsymbol{w}^\top(\boldsymbol{x}'-\boldsymbol{x}))\mathbb{I}_{[y=-1\wedge y'=1]},$$

where

$$\mathbb{I}_{[y=1\wedge y'=-1]} = \begin{cases} 1 & \text{if } y=1 \text{ and } y'=-1, \\ 0 & \text{otherwise.} \end{cases}$$

and similarly for $\mathbb{I}_{[y=-1\wedge y'=1]}$. Also, $\ell$ is a surrogate loss function, e.g., the hinge loss

$$\ell(t) = (1-t)_+ = \begin{cases} 0 & \text{if } 1-t < 0, \\ 1-t & \text{otherwise.} \end{cases}$$

---

[6] In (Yang et al., 2021b), the authors have shown that OGD can remarkably outperform existing algorithms, including OLP (Kar et al., 2013), OAM$_{\text{gra}}$ (Zhao et al., 2011), SGD$_{\text{pair}}$ (Lei et al., 2020), and SPAUC (Lei & Ying, 2021).

[7] $[-5:5]$ stands for integers in the interval $[-5,5]$.

**Remark 2** *The loss function above follows the setup in (Yang et al., 2021b) which is designed for the AUC maximization problem with data stream $(\boldsymbol{x}, y)$ sampled from a distribution that contains both positive and negative samples by utilizing the indicator function $\mathbb{I}_{[y=1 \wedge y'=-1]}$ and $\mathbb{I}_{[y=-1 \wedge y'=1]}$. We want to mention that the second term is not presented in the main text of (Yang et al., 2021b) but implicitly utilized their code [8] on which our implementation is based. In particular, in both OGD and AOGD, the loss is activated or non-zero whenever the consecutive sample pairs $(\boldsymbol{x}, y)$ and $(\boldsymbol{x}', y')$ are of opposite labels, i.e., $y = 1$ and $y' = -1$ or $y = -1$ and $y' = 1$.*

*We now provide further clarification on the pairwise model and AUC maximization: In model (1), the distribution for $\xi$ and $\xi'$ is assumed to be the same. This is in contrast to the early AUC maximization formulation, as seen in (Zhao et al., 2011), which does not have this property. In the early formulation, the positive and negative classes are treated differently, with potentially different distributions. However, we can introduce indicator functions and show that the AUC maximization can be represented as a special form of (1) as above.*

*A natural question that arises is why we would choose to reformulate the AUC maximization problem into the model (1). At first glance, there does not seem to be any apparent advantage to such a reformulation, as the algorithm only works when a negative sample is encountered along with a positive sample. In the worst-case scenario, if all the negative labels are encountered before the positive labels, the algorithm would almost output no predictions.*

*We provide a necessary explanation here:*

- *a) The worst-case scenario mentioned above is highly unlikely to occur because we assume that the data is independently and identically distributed from the underlying distribution.*

- *b) In the online setting, the labels are unknown in advance. While the work of Zhao et al. (2011) discusses the online scenario, they employ an "update buffer" data pre-processing step to divide the data and subsequently run the algorithm offline. Therefore, we cannot directly apply the formulation described in (Zhao et al., 2011). To address this, it is more flexible to utilize the formulation presented in (Yang et al., 2021b). This formulation is better suited for online learning, as it does not require an offline processing step like the one used in (Zhao et al., 2011).*

- *c) Existing results from (Yang et al., 2021b) demonstrate that the formulation (1) outperforms previous methods, even when a negative sample is encountered along with a positive sample. This is because previous methods often have high gradient complexity at each iteration, while the online complexity is very small in the proposed formulation.*

*In summary, the reformulation (1) allows for more flexibility in the optimization process and has the potential to yield better results in practical scenarios.*

Figure 2 plots the AUC scores of AOGD, OGD, and AdaOAM against the number of iterations (in log scale) [9]

Table 2: Average AUC scores $\pm$ standard deviation with convex loss function across the six benchmark datasets listed in Table 1. The best results are highlighted in boldface.

| Algorithm | diabetes | german | ijcnn1 | letter | mnist | usps |
|---|---|---|---|---|---|---|
| AOGD | $\mathbf{.831 \pm .027}$ | $\mathbf{.795 \pm .026}$ | $\mathbf{.934 \pm .002}$ | $\mathbf{.814 \pm .006}$ | $.931 \pm .002$ | $.925 \pm .004$ |
| OGD | $.831 \pm .030$ | $.793 \pm .021$ | $\mathbf{.934 \pm .002}$ | $.810 \pm .007$ | $\mathbf{.932 \pm .001}$ | $\mathbf{.926 \pm .006}$ |
| AdaOAM | $.829 \pm .027$ | $.792 \pm .028$ | $.934 \pm .003$ | $.814 \pm .009$ | $.932 \pm .002$ | $.924 \pm .005$ |

Table 2 summarizes the generalization performance between AOGD and OGD. The results for AOGD and AdaOAM are obtained from our above experiment and the results for OGD are adapted from (Yang et al.,

---

[8]https://github.com/zhenhuan-yang/simple-pairwise

[9]Number of iterations $N$ is given by $2^{N_{log}/2+4}$ where $N_{log}$ is the log-scaled number of iterations. on the six benchmark datasets listed in Table 1. The numerical results on the six benchmark datasets show that AOGD converges faster than OGD and AdaOAM in general, confirming our theoretical results.

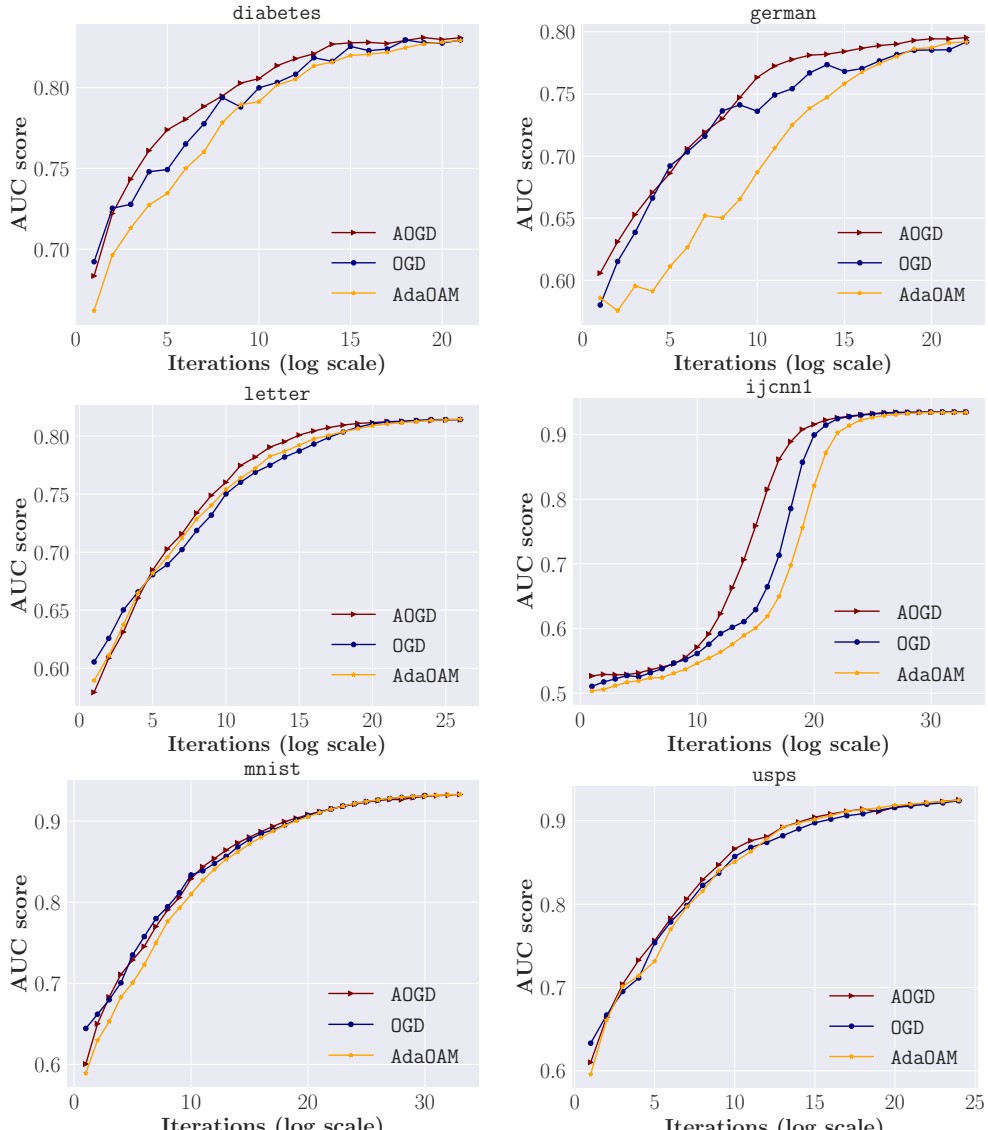

Figure 2: The AUC score of AOGD and OGD against the number of iterations (in log scale) for AUC maximization with **convex loss** (hinge loss). It is evident that AOGD converges faster than OGD and AdaOAM in general, confirming our established theoretical results for AOGD.

2021b). Overall, AOGD generalizes as well as OGD and AdaOAM as the peak performance of these three methods are comparable. Establishing the generalization of OAGD is an interesting future direction.

**Non-convex case:** We follow the setting in Appendix G of (Yang et al., 2021b) on AUC maximization using the logistic link function $\text{logit}(t) = (1 + \exp(-t))^{-1}$ and then the square loss function $\ell(t) = (1 - t)^2$. Hence, the loss function for the AUC maximization problem is given by the non-convex function

$$f(\boldsymbol{w}; (\boldsymbol{x}, y), (\boldsymbol{x}', y')) = \left(1 - \text{logit}\left(\mathbf{w}^\top (\mathbf{x} - \mathbf{x}')\right)\right)^2 \mathbb{I}_{[y=1 \wedge y'=-1]} + \left(1 - \text{logit}\left(\mathbf{w}^\top (\mathbf{x}' - \mathbf{x})\right)\right)^2 \mathbb{I}_{[y=-1 \wedge y'=1]}.$$

We plot in Figure 3 the AUC scores of AOGD, OGD, and AdaOAM against the number of iterations on the six benchmark datasets listed in Table 1. The numerical results on the six benchmark datasets show that AOGD converges faster than OGD and AdaOAM in general, confirming our theoretical results.

Table 3: Average AUC scores $\pm$ standard deviation with nonconvex loss function across the six benchmark datasets listed in Table 1. The best results are highlighted in boldface.

| Algorithm | diabetes | german | ijcnn1 | letter | mnist | usps |
|---|---|---|---|---|---|---|
| AOGD | $\mathbf{.834 \pm .022}$ | $\mathbf{.797 \pm .023}$ | $\mathbf{.935 \pm .002}$ | $.815 \pm .005$ | $\mathbf{.932 \pm .002}$ | $\mathbf{.928 \pm .004}$ |
| OGD | $.829 \pm .033$ | $.794 \pm .022$ | $.934 \pm .002$ | $\mathbf{.815 \pm .003}$ | $.931 \pm .002$ | $.926 \pm .005$ |
| AdaOAM | $.831 \pm .025$ | $.789 \pm .021$ | $.931 \pm .002$ | $.815 \pm .006$ | $.932 \pm .003$ | $.926 \pm .007$ |

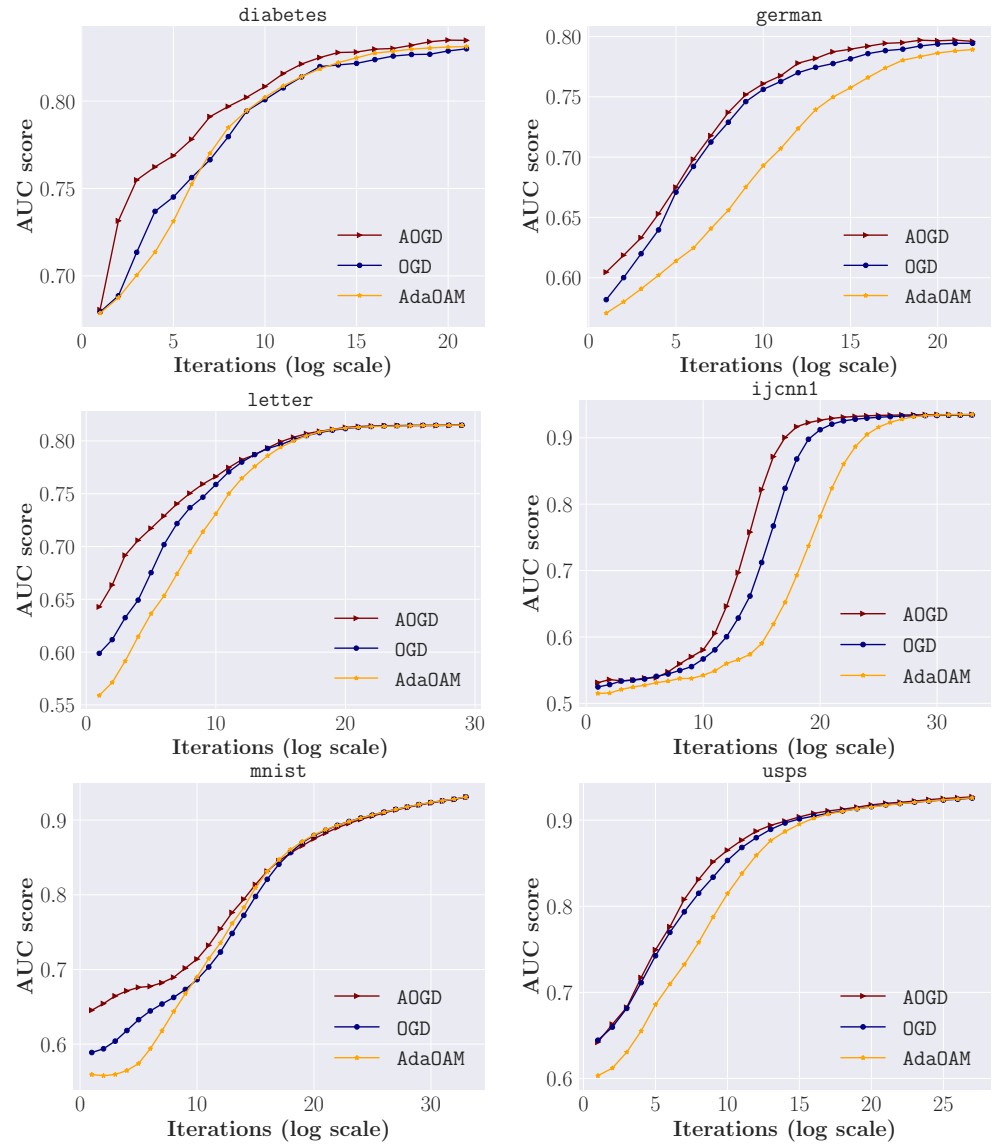

Figure 3: The AUC score of AOGD, OGD, and AdaOAM against the number of iterations (in log scale) with non-convex loss function. In this case, AOGD converges faster than OGD and AdaOAM in general, confirming our established theoretical results for AOGD. In particular, the slower convergence of AdaOAM is more prominent in the non-convex case.

# 4 More Related Works

Because offline methods employ the ERM-like policy, the core problem of most offline methods is establishing the generalization bound of the finite-sum model with statistical learning theory or algorithmic stability

(Agarwal & Niyogi, 2009; Jin et al., 2009; Wang et al., 2019; Gao & Zhou, 2013; Lei et al., 2020). It is worth mentioning that the difficulty in the generalization analysis for pairwise learning lies in that the objective functions fail to be i.i.d. with each other, which breaks the fundamental assumption in statistical learning theory and the algorithmic stability communities.

The online methods for pairwise learning assume the model accesses a data stream of i.i.d. samples, including online AUC maximization algorithms (Zhao et al., 2011; Ying et al., 2016; Liu et al., 2018; Natole et al., 2018; Lei & Ying, 2021; Guo et al., 2020), online metric learning algorithms (Shalev-Shwartz et al., 2004; Davis et al., 2007; Jain et al., 2008; Jin et al., 2009), online learning to rank (Rejchel, 2012; Schuth et al., 2013; Zoghi et al., 2017; Li et al., 2019), neural link prediction (Wang et al., 2021), etc. The online AUC maximization is proposed by Zhao et al. (2011) with theoretical guarantees. In the paper (Ying et al., 2016), a stochastic online AUC maximization algorithm is proposed from the perspective of a saddle representation. The main advantage of the algorithm in (Ying et al., 2016) is to avoid storing all previous examples and second-order covariance matrices. Leveraging saddle representation, Liu et al. (2018) propose a faster online AUC maximization algorithm with provably improved statistical convergence rates. The stochastic proximal algorithms for AUC maximization with non-differentiable regularization are proposed and studied in (Natole et al., 2018). To make the algorithm scalable to large-scale streaming data, Lei & Ying (2021) propose a new stochastic proximal algorithm. In the paper (Guo et al., 2020), the authors consider the distributed setting and propose a communication-efficient stochastic AUC maximization with deep neural networks. Shalev-Shwartz et al. (2004) propose an online algorithm for supervised learning of pseudo-metrics. In (Davis et al., 2007), the authors present an information-theoretic approach for online metric learning. In (Jain et al., 2008), leveraging the LogDet regularization, the authors propose a fast online metric learning for the similarity search. The generalization bound of regularized distance metric learning is established in (Jin et al., 2009). Rejchel (2012) consider ranking estimators that minimize the convex empirical risks and prove their generalization bounds. A framework of online learning to rank is proposed by Schuth et al. (2013). In the paper (Zoghi et al., 2017), the authors investigate online learning to rank in stochastic click models. Paper (Li et al., 2019) introduces a new model for online ranking with features. The differentially private pairwise learning has been recently studied, and representative works include (Huai et al., 2020; Yang et al., 2021a; Xue et al., 2021; Yang et al., 2021b).

## 5 Conclusions

In this paper, we propose adaptive online gradient descent algorithms to solve pairwise learning problems and establish their theoretical performance bounds in strongly convex, convex, and nonconvex settings. Our theoretical results explain why the convergence speed of adaptive online gradient descent can outperform the one without adaptive stepsize for pairwise learning. We also provide numerical experiments to demonstrate the efficiency of the proposed algorithm.

**Limitation and future work.** There are two major limitations in our analysis: 1) we assume the set $\mathcal{K}$ is bounded in establishing Theorem 1, and 2) the convergence rate in Theorem 2 is analyzed for the adaptive online gradient descent without momentum. We leave how to overcome the above two limitations as future work. There are numerous other avenues for future work, including 1) Can we establish the lower bound of the convergence rates for the adaptive online gradient descent applied to pairwise learning? 2) Can we extend the online adaptive gradient descent to the proximal settings to solve nonsmooth pairwise learning problems?

**Broader Impact Statement**

N/A

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

# A Proofs of Results in the Convex Scenario

## A.1 Technical Lemmas

Given any $\boldsymbol{x}^* \in \arg_{\mathcal{K}} \min f$, in the (strongly) convex setting, we introduce the following notation

$$
\begin{cases}
\phi_k := \mathbb{E}\left[\frac{\|\boldsymbol{g}^k\|^2}{\sqrt{v^k}}\right] + 2B^2\mathbb{E}\left[\frac{1}{\sqrt{v^{k-2}}} - \frac{1}{\sqrt{v^{k-1}}}\right] + \frac{L^2\mathbb{E}\|\boldsymbol{x}^k-\boldsymbol{x}^{k-1}\|^2}{\sqrt{\delta}} + B^2\mathbb{E}\left[\frac{1}{\sqrt{v^{k-2}}} - \frac{1}{\sqrt{v^k}}\right], \\
\qquad\qquad A_k := \mathbb{E}(\|\boldsymbol{m}^k\|^2/(v^k)^{\frac{1}{2}}), \\
\qquad\qquad B_k := \mathbb{E}\left(\langle \boldsymbol{x}^* - \boldsymbol{x}^k, \boldsymbol{m}^k\rangle\right), \\
\qquad C_k := (\eta\theta + \frac{1-\theta}{2})A_{k-1} + (1-\theta)\phi_k
\end{cases} \tag{9}
$$

We have the following lemmas.

**Lemma 1** *[Lemma 9 in the appendix, (Li & Orabona, 2019)] Let $a_1, a_2, \ldots, a_K$ be non-negative, and $h$ be a non-increasing function. Then we have*

$$
\sum_{k=1}^{K} a_k h(a_0 + \sum_{i=1}^{k} a_i) \leq \int_{a_0}^{\sum_{k=0}^{K} a_k} h(t)dt.
$$

**Lemma 2** *Assume $\{\boldsymbol{x}^k\}_{k\geq 1}$ is generated by Algorithm 2, then we have*

$$
\sum_{k=1}^{K} A_k \leq \sum_{k=1}^{K} \mathbb{E}[\|\boldsymbol{g}^k\|^2/(v^k)^{\frac{1}{2}}] \leq \mathbb{E}(v^k)^{\frac{1}{2}}.
$$

**Lemma 3** *Assume $\{\boldsymbol{x}^k\}_{k\geq 1}$ is generated by Algorithm 2 and $\|\boldsymbol{g}^0\| \geq \sqrt{\delta} > 0$, then we have*

$$
\sum_{k=1}^{K} \mathbb{E}\|\boldsymbol{x}^k - \boldsymbol{x}^{k-1}\|^2 \leq \sum_{k=1}^{K} \mathbb{E}[\|\boldsymbol{g}^k\|^2/v^k] \leq \mathbb{E}\ln\frac{v^k}{\delta}.
$$

**Lemma 4** *Assume $\{\boldsymbol{x}^k\}_{k\geq 1}$ is generated by Algorithm 2, then we have*

$$
\mathbb{E}\|\nabla F(\boldsymbol{x}^{k-1}; \xi^k, \xi^{k-1})\|^2 \leq 2L^2\mathbb{E}\|\boldsymbol{x}^k - \boldsymbol{x}^{k-1}\|^2 + 2\mathbb{E}\|\boldsymbol{g}^k\|^2.
$$

**Lemma 5** *Assume $\{\boldsymbol{x}^k\}_{k\geq 1}$ is generated by Algorithm 2, and Assumptions 1 and 2 hold, then the following result holds*

$$
B_k \leq \theta B_{k-1} + C_k + (1-\theta)\mathbb{E}\left[f(\boldsymbol{x}^*) - f(\boldsymbol{x}^k)\right].
$$

## A.2 Proof of Theorem 1

Given $k \in \mathbb{Z}^+$, with Lemma 5 and mathematical induction, we have the following inequality

$$
B_k \leq \theta^k B_1 + \sum_{i=1}^{k-1} \theta^{k-1-i} C_i + \sum_{i=1}^{k-1} (1-\theta)\theta^{k-1-i}\mathbb{E}(f(\boldsymbol{x}^*) - f(\boldsymbol{x}^i)).
$$

Notice that $\boldsymbol{m}^1 = \boldsymbol{0}$ and $B_1 = 0$, we get

$$
B_k \leq \sum_{i=1}^{k-1} \theta^{k-1-i} C_i + \sum_{i=1}^{k-1} (1-\theta)\theta^{k-1-i}\mathbb{E}(f(\boldsymbol{x}^*) - f(\boldsymbol{x}^i)). \tag{10}
$$

Summing the inequality (10) from $k = 1$ to $K$ gives us

$$\sum_{k=1}^{K} B_k \leq \sum_{k=1}^{K} \sum_{i=1}^{k-1} \theta^{k-1-i} C_i + \sum_{k=1}^{K} \sum_{i=1}^{k-1} (1-\theta) \theta^{k-1-i} \mathbb{E}(f(\boldsymbol{x}^*) - f(\boldsymbol{x}^i))$$

$$\leq \frac{\sum_{k=1}^{K} C_k}{1-\theta} + (1-\theta) \sum_{k=1}^{K} \mathbb{E}(f(\boldsymbol{x}^*) - f(\boldsymbol{x}^k)).$$

Therefore, we have

$$\sum_{k=1}^{K} \mathbb{E}(f(\boldsymbol{x}^k) - f(\boldsymbol{x}^*)) \leq -\sum_{k=1}^{K} \frac{B_k}{(1-\theta)} + \frac{\sum_{k=1}^{K} C_k}{(1-\theta)^2}. \tag{11}$$

The scheme of the algorithm indicates that

$$\boldsymbol{x}^{k+1} = \mathbf{Proj}_{\mathcal{K}}(\boldsymbol{x}^k - \eta \boldsymbol{m}^k / (v^k)^{\frac{1}{2}}).$$

We then get

$$\|\boldsymbol{x}^{k+1} - \boldsymbol{x}^*\|^2 = \|\mathbf{Proj}_{\mathcal{K}}(\boldsymbol{x}^k - \eta \boldsymbol{m}^k / (v^k)^{\frac{1}{2}}) - \boldsymbol{x}^*\|^2$$

$$= \|\mathbf{Proj}_{\mathcal{K}}(\boldsymbol{x}^k - \eta \boldsymbol{m}^k / (v^k)^{\frac{1}{2}}) - \mathbf{Proj}_{\mathcal{K}}(\boldsymbol{x}^*)\|^2$$

$$\leq \|\boldsymbol{x}^k - \eta \boldsymbol{m}^k / (v^k)^{\frac{1}{2}} - \boldsymbol{x}^*\|^2.$$

Multiplying both sides with $(v^k)^{\frac{1}{2}}$, we are then led to

$$(v^k)^{\frac{1}{2}} \|\boldsymbol{x}^{k+1} - \boldsymbol{x}^*\|^2 \leq (v^k)^{\frac{1}{2}} \|\boldsymbol{x}^* - \boldsymbol{x}^k\|^2 + 2\eta \langle \boldsymbol{m}^k, \boldsymbol{x}^* - \boldsymbol{x}^k \rangle + \eta^2 \|\boldsymbol{m}^k\|^2 / (v^k)^{\frac{1}{2}},$$

which is equivalent to

$$(v^{k+1})^{\frac{1}{2}} \|\boldsymbol{x}^{k+1} - \boldsymbol{x}^*\|^2 \leq (v^k)^{\frac{1}{2}} \|\boldsymbol{x}^* - \boldsymbol{x}^k\|^2 + 2\eta \langle \boldsymbol{m}^k, \boldsymbol{x}^* - \boldsymbol{x}^k \rangle$$

$$+ \eta^2 \|\boldsymbol{m}^k\|^2 / (v^k)^{\frac{1}{2}} + ((v^{k+1})^{\frac{1}{2}} - (v^k)^{\frac{1}{2}}) \|\boldsymbol{x}^{k+1} - \boldsymbol{x}^*\|^2.$$

Taking the total expectation of both sides of the above equation gives us

$$-2\eta B_k \leq \mathbb{E}(v^k)^{\frac{1}{2}} \|\boldsymbol{x}^k - \boldsymbol{x}^*\|^2 - \mathbb{E}(v^{k+1})^{\frac{1}{2}} \|\boldsymbol{x}^{k+1} - \boldsymbol{x}^*\|^2$$

$$+ \eta^2 A_k + (\mathbb{E}(v^{k+1})^{\frac{1}{2}} - \mathbb{E}(v^k)^{\frac{1}{2}}) D^2. \tag{12}$$

Summing (12) from $k = 1$ to $K$ gives us

$$\sum_{k=1}^{K} (-B_k)/(1-\theta) \leq \frac{\mathbb{E}\sqrt{v^1} \|\boldsymbol{x}^* - \boldsymbol{x}^1\|^2}{2\eta(1-\theta)} + \frac{\eta}{2(1-\theta)} \sum_{k=1}^{K} A_k + \frac{D^2}{(1-\theta)} \mathbb{E}\sqrt{v^{K+1}}.$$

Together with (11), we then have

$$\sum_{k=1}^{K} \mathbb{E}(f(\boldsymbol{x}^k) - f(\boldsymbol{x}^*)) \leq \frac{\mathbb{E}\sqrt{v^1} \|\boldsymbol{x}^* - \boldsymbol{x}^1\|^2}{2\eta(1-\theta)}$$

$$+ \frac{\eta}{2(1-\theta)} \sum_{k=1}^{K} A_k + \frac{D^2}{(1-\theta)} \mathbb{E}\sqrt{v^{K+1}} + \frac{\sum_{k=1}^{K} C_k}{(1-\theta)^2}. \tag{13}$$

We turn to bound the right-hand side of (13) and get the following bound

$$\frac{\eta}{2(1-\theta)} \sum_{k=1}^{K} A_k \leq \frac{\eta}{2(1-\theta)} \mathbb{E}(v^{K+1})^{\frac{1}{2}}. \tag{14}$$

On the other hand, we can get

$$\frac{\sum_{k=1}^K C_k}{(1-\theta)^2} \leq \frac{\eta\theta}{(1-\theta)^2}\mathbb{E}(v^{K+1})^{\frac{1}{2}} + \frac{2L}{(1-\theta)^2}\mathbb{E}\ln\frac{v^{K+1}}{\delta} + \frac{1}{(1-\theta)^2}\sum_{k=1}^K \mathbb{E}\phi_k$$

$$\leq \frac{\eta\theta}{(1-\theta)^2}\mathbb{E}(v^{K+1})^{\frac{1}{2}} + \frac{2L}{(1-\theta)^2}\mathbb{E}\ln\frac{v^{K+1}}{\delta} + \frac{\frac{3B^2}{\sqrt{\delta}} + \frac{L^2}{\sqrt{\delta}}\mathbb{E}\ln\frac{v^{K+1}}{\delta} + \mathbb{E}(v^{K+1})^{\frac{1}{2}}}{(1-\theta)^2} \tag{15}$$

Substituting the bounds (14) and (15) into (13), we are then led to

$$\Big[\sum_{k=1}^K \mathbb{E}(f(\boldsymbol{x}^k) - f(\boldsymbol{x}^*))\Big]$$

$$\leq \frac{\mathbb{E}\sqrt{v^1}\|\boldsymbol{x}^* - \boldsymbol{x}^1\|^2}{2\eta(1-\theta)} + \Big[\frac{\eta + 2D^2}{2(1-\theta)} + \frac{\eta\theta + 1}{(1-\theta)^2}\Big]\cdot\mathbb{E}\sqrt{v^{K+1}} \tag{16}$$

$$+ \Big[\frac{4L}{(1-\theta)^2} + \frac{2L^2}{(1-\theta)^2\sqrt{\delta}}\Big]\mathbb{E}\ln\frac{\sqrt{v^{K+1}}}{\sqrt{\delta}} + \frac{3B^2}{(1-\theta)^2\sqrt{\delta}}.$$

Notice that $-\ln(\cdot)$ and $-\frac{1}{\cdot}$ are both convex when $\cdot$ is positive, using Jensen's inequality gives us

$$-\ln\mathbb{E}\sqrt{v^{K+1}} \leq -\mathbb{E}\ln\sqrt{v^{K+1}}, \quad -\frac{1}{\mathbb{E}(\sqrt{v^k})} \leq -\mathbb{E}\frac{1}{\sqrt{v^k}}$$

$$\Rightarrow \mathbb{E}\ln\sqrt{v^{K+1}} \leq \ln\mathbb{E}\sqrt{v^{K+1}}, \quad \mathbb{E}\frac{1}{\sqrt{v^k}} \leq \frac{1}{\mathbb{E}\sqrt{v^k}}.$$

Therefore, (16) can be presented as

$$\mathbb{E}\Big[f\Big(\frac{\sum_{k=1}^K \boldsymbol{x}^k}{K}\Big) - \min f\Big] \leq \frac{c_1 + c_2(K)}{K},$$

where

$$c_1 := \frac{\mathbb{E}\sqrt{v^1}\|\boldsymbol{x}^* - \boldsymbol{x}^1\|^2}{2(1-\theta)\eta} + \frac{3B^2}{(1-\theta)^2\sqrt{\delta}},$$

and

$$c_2(K) := \Big[\frac{\eta + 2D^2}{2(1-\theta)} + \frac{\eta\theta + 1}{(1-\theta)^2}\Big]\cdot\mathbb{E}\sqrt{v^{K+1}} + \Big[\frac{4L}{(1-\theta)^2} + \frac{2L^2}{(1-\theta)^2\sqrt{\delta}}\Big]\ln\frac{\mathbb{E}\sqrt{v^{K+1}}}{\sqrt{\delta}}.$$

## B  Proofs of Results in the Strongly Convex Scenario

### B.1  Proof of Proposition 2

The boundedness of the gradient and the strong convexity indicate $\mathcal{K}$ is bounded, whose radius is assumed to be $D > 0$. Notice that the operator $\mathbf{Proj}_{\mathcal{K}}(\cdot)$ is contractive, as $\theta = 0$, we get

$$\mathbb{E}\|\boldsymbol{x}^{k+1} - \boldsymbol{x}^*\|^2 = \mathbb{E}\|\mathbf{Proj}_{\mathcal{K}}(\boldsymbol{x}^k - \frac{\eta}{\sqrt{k}}\boldsymbol{m}^k/\sqrt{v^k}) - \boldsymbol{x}^*\|^2$$

$$= \mathbb{E}\|\mathbf{Proj}_{\mathcal{K}}(\boldsymbol{x}^k - \frac{\eta}{\sqrt{k}}\boldsymbol{m}^k/\sqrt{v^k}) - \mathbf{Proj}_{\mathcal{K}}(\boldsymbol{x}^*)\|^2$$

$$\leq \mathbb{E}\|\boldsymbol{x}^k - \frac{\eta}{\sqrt{k}}\boldsymbol{m}^k/\sqrt{v^k} - \boldsymbol{x}^*\|^2$$

$$= \mathbb{E}\|\boldsymbol{x}^k - \boldsymbol{x}^*\|^2 - 2\frac{\eta}{\sqrt{k}}\mathbb{E}\big(\frac{\langle\nabla F(\boldsymbol{x}^k;\xi^k,\xi^{k-1}),\boldsymbol{x}^k - \boldsymbol{x}^*\rangle}{\sqrt{v^k}}\big) + \frac{\eta^2}{k}\mathbb{E}\|\boldsymbol{m}^k\|^2/v^k$$

$$\leq \mathbb{E}\Big[(1 - 2\eta\nu/\sqrt{kv^k})\mathbb{E}\|\boldsymbol{x}^k - \boldsymbol{x}^*\|^2\Big] + \frac{\eta^2}{k}\mathbb{E}\frac{\|\boldsymbol{m}^k\|^2}{v^k}$$

$$+ 2\frac{\eta}{\sqrt{k}}\mathbb{E}([F(\boldsymbol{x}^*;\xi^k,\xi^{k-1}) - F(\boldsymbol{x}^k;\xi^k,\xi^{k-1})]/\sqrt{v^k}),$$

where we used the strong convexity of $F(\boldsymbol{x}; \xi^k, \xi^{k-1})$. Now, we turn to bound

$$\mathbb{E}([F(\boldsymbol{x}^*; \xi^k, \xi^{k-1}) - F(\boldsymbol{x}^k; \xi^k, \xi^{k-1})]/\sqrt{v^k}).$$

With direct computations, we have the decomposition

$$
\begin{aligned}
&\mathbb{E}([F(\boldsymbol{x}^*; \xi^k, \xi^{k-1}) - F(\boldsymbol{x}^k; \xi^k, \xi^{k-1})]/\sqrt{v^k}) \\
&= \mathbb{E}([F(\boldsymbol{x}^*; \xi^k, \xi^{k-1}) - F(\boldsymbol{x}^{k-1}; \xi^k, \xi^{k-1})]/\sqrt{v^{k-2}}) \\
&\quad + \mathbb{E}([F(\boldsymbol{x}^{k-1}; \xi^k, \xi^{k-1}) - F(\boldsymbol{x}^k; \xi^k, \xi^{k-1})]/\sqrt{v^{k-2}}) \\
&\quad + \frac{\eta}{\sqrt{k}}\mathbb{E}\Big([F(\boldsymbol{x}^*; \xi^k, \xi^{k-1}) - F(\boldsymbol{x}^k; \xi^k, \xi^{k-1})] \times (1/\sqrt{v^k} - 1/\sqrt{v^{k-2}})\Big),
\end{aligned}
$$

consequently,

$$
\begin{aligned}
&\mathbb{E}([F(\boldsymbol{x}^*; \xi^k, \xi^{k-1}) - F(\boldsymbol{x}^k; \xi^k, \xi^{k-1})]/\sqrt{v^k}) \\
&\le \mathbb{E}\Big(\frac{\langle \boldsymbol{m}^{k-1}, \nabla F(\boldsymbol{x}^{k-1}; \xi^k, \xi^{k-1}) \rangle}{\sqrt{k-1}\sqrt{v^{k-1}}\sqrt{v^{k-2}}}\Big) + \frac{DB\eta}{\sqrt{k}}\mathbb{E}(1/\sqrt{v^{k-2}} - 1/\sqrt{v^k}) \\
&\le \frac{1}{2\sqrt{k-1}}\mathbb{E}\frac{\|\boldsymbol{m}^{k-1}\|}{v^{k-1}} + \frac{1}{2\sqrt{k-1}}\mathbb{E}\frac{\|\nabla F(\boldsymbol{x}^{k-1}; \xi^k, \xi^{k-1})\|^2}{v^{k-2}} \\
&\quad + \frac{DB\eta}{\sqrt{k}}\mathbb{E}(1/\sqrt{v^{k-2}} - 1/\sqrt{v^k}) + \frac{B}{\sqrt{k-1}\sqrt{\delta}}\mathbb{E}(1/\sqrt{v^{k-2}} - 1/\sqrt{v^{k-1}}) \\
&\le \frac{1}{\sqrt{\delta}}\mathbb{E}\Big(\frac{\|\boldsymbol{m}^{k-1}\|^2}{\sqrt{k-1}v^{k-1}}\Big) + \frac{DB\eta}{\sqrt{k}}\mathbb{E}(1/\sqrt{v^{k-2}} - 1/\sqrt{v^k}) \\
&\quad + \frac{B}{\sqrt{k-1}\sqrt{\delta}}\mathbb{E}(1/\sqrt{v^{k-2}} - 1/\sqrt{v^{k-1}}).
\end{aligned}
$$

where we used $\mathbb{E}([F(\boldsymbol{x}^*; \xi^k, \xi^{k-1}) - F(\boldsymbol{x}^{k-1}; \xi^k, \xi^{k-1})]/\sqrt{v^{k-2}}) = [f(\boldsymbol{x}^*) - f(\boldsymbol{x}^{k-1})]/\sqrt{v^{k-2}} \le 0$. Letting $\eta = \frac{B}{2\nu}$ and $\beta_k := \frac{\eta}{\sqrt{k}\sqrt{k-1}}\mathbb{E}\frac{\|\boldsymbol{m}^{k-1}\|}{v^{k-1}} + \frac{\eta}{\sqrt{k}\sqrt{k-1}}\mathbb{E}\frac{\|\nabla F(\boldsymbol{x}^{k-1}; \xi^k, \xi^{k-1})\|^2}{v^{k-2}} + \frac{2DB\eta^2}{k}\mathbb{E}(1/\sqrt{v^{k-2}} - 1/\sqrt{v^k}) + \frac{2B\eta}{\sqrt{k-1}\sqrt{k}\sqrt{\delta}}\mathbb{E}(1/\sqrt{v^{k-2}} - 1/\sqrt{v^{k-1}}) + \frac{\eta^2}{k}\mathbb{E}\frac{\|\boldsymbol{m}^k\|^2}{v^k}$,

$$a_{k+1} \le (1 - \frac{1}{k})a_k + \beta_k.$$

With direct computations, we have

$$
\begin{aligned}
a_3 &\le \frac{1}{2}a_2 + \beta_2, \\
a_4 &\le \frac{2}{3}a_3 + \beta_3 \le \frac{1}{3}a_2 + \frac{2}{3}\beta_2 + \beta_3, \\
a_5 &\le \frac{3}{4}a_4 + \beta_4 \le \frac{1}{4}a_2 + \frac{2}{4}\beta_2 + \frac{3}{4}\beta_3 + \beta_4, \\
&\vdots \\
a_{K+1} &\le \frac{a_2}{K} + \sum_{k=2}^{K}\frac{k\beta_k}{K}.
\end{aligned}
$$

Notice that

$$
\begin{aligned}
k\beta_k = \mathcal{O}\Big(&\mathbb{E}\frac{\|\boldsymbol{m}^{k-1}\|}{v^{k-1}} + \mathbb{E}\frac{\|\nabla F(\boldsymbol{x}^{k-1}; \xi^k, \xi^{k-1})\|^2}{v^{k-2}} + \mathbb{E}(1/\sqrt{v^{k-2}} - 1/\sqrt{v^k}) \\
&+ \mathbb{E}(1/\sqrt{v^{k-2}} - 1/\sqrt{v^{k-1}}) + \mathbb{E}\frac{\|\boldsymbol{m}^k\|^2}{v^k}\Big),
\end{aligned}
$$

we just need to compute $\sum_{k=1}^{\infty} \mathbb{E} \frac{\|\nabla F(\boldsymbol{x}^{k-1}; \xi^k, \xi^{k-1})\|^2}{v^{k-2}}$. By using Lemma 4, we have

$$\mathbb{E}\|\nabla F(\boldsymbol{x}^{k-1}; \xi^k, \xi^{k-1})\|^2 / v^{k-2} \leq 2L^2 \mathbb{E}\|\boldsymbol{x}^k - \boldsymbol{x}^{k-1}\|^2 / \delta + 2\mathbb{E}\|\boldsymbol{g}^k\|^2 / v^{k-2}.$$

With the fact

$$\mathbb{E}\frac{\|\boldsymbol{g}^k\|^2}{v^{k-2}} \leq \mathbb{E}\frac{\|\boldsymbol{g}^k\|^2}{v^k} + B^2(\mathbb{E}1/v^{k-2} - \mathbb{E}1/v^k),$$

we then get

$$\sum_{k=1}^{\infty} \mathbb{E}\frac{\|\nabla F(\boldsymbol{x}^{k-1}; \xi^k, \xi^{k-1})\|^2}{v^{k-2}} = \mathcal{O}(\ln v^K).$$

With Lemma 3, we can get

$$\sum_{k=2}^{K} k\beta_k = \mathcal{O}(\ln v^K) = \mathcal{O}(\ln K) \Rightarrow a_{K+1} = \frac{\ln K}{K}.$$

## C   Proofs of Results in Nonconvex Scenario

### C.1   Additional Technical Lemmas

In the nonconvex case, we denote the following items to simplify the presentations of the technical lemmas

$$\begin{cases} \hat{A}_k := \mathbb{E}\Big[\|\boldsymbol{m}^k\|^2 / v^k\Big], \\ \hat{B}_k := \mathbb{E}\left(\langle -\nabla f(\boldsymbol{x}^k), \boldsymbol{m}^k/(v^k)^{\frac{1}{2}}\rangle\right), \\ \hat{C}_k := \theta\eta\hat{A}_k + 2(1-\theta)\mathbb{E}\Big[\|\boldsymbol{g}^k\|^2/v^k\Big] + 6(1-\theta)B^2\mathbb{E}[1/(v^{k-2})^{\frac{1}{2}} - 1/(v^k)^{\frac{1}{2}}] \\ \qquad + (1-\theta)(2L^2/\sqrt{\delta} + L^2/\delta + 1/2)\mathbb{E}\|\boldsymbol{x}^k - \boldsymbol{x}^{k-1}\|^2. \end{cases} \tag{17}$$

**Lemma 6**  *Assume $\{\boldsymbol{x}^k\}_{k\geq 1}$ is generated by Algorithm 2, then we have*

$$\sum_{k=1}^{K} \hat{A}_k \leq \sum_{k=1}^{K} \mathbb{E}\|\boldsymbol{g}^k\|^2 / v^k \leq \mathbb{E}\ln\frac{v^k}{\delta}.$$

**Lemma 7**  *Assume $\{\boldsymbol{x}^k\}_{k\geq 1}$ is generated by Algorithm 2 and the functions are nonconvex. Let $\mathcal{K}$ be the full space and Assumptions 1 and 2 hold, then the following result holds*

$$\hat{B}_k + \frac{(1-\theta)}{2}\mathbb{E}\Big(\|\nabla f(\boldsymbol{x}^k)\|^2/(v^k)^{\frac{1}{2}}\Big) \leq \theta\hat{B}_{k-1} + \hat{C}_k.$$

### C.2   Proof of Theorem 3

According to Lemma 7, we have

$$\frac{(1-\theta)}{2}\sum_{k=1}^{K} \mathbb{E}\Big(\|\nabla f(\boldsymbol{x}^k)\|^2/(v^k)^{\frac{1}{2}}\Big) \leq -\hat{B}_K + (\theta-1)\sum_{k=1}^{K-1} \hat{B}_k + \sum_{k=1}^{K} \hat{C}_k$$

$$\leq (\theta-1)\sum_{k=1}^{K-1} \hat{B}_k + \sum_{k=1}^{K} \hat{C}_k + \frac{B^2}{\sqrt{\delta}}. \tag{18}$$

The Lipschitz property of the gradients gives

$$\mathbb{E}f(\boldsymbol{x}^{k+1}) - \mathbb{E}f(\boldsymbol{x}^k) \leq \mathbb{E}\langle \nabla f(\boldsymbol{x}^k), \boldsymbol{x}^{k+1} - \boldsymbol{x}^k\rangle + \frac{L\mathbb{E}\|\boldsymbol{x}^{k+1} - \boldsymbol{x}^k\|^2}{2} = \eta\hat{B}_k + \frac{L\eta^2}{2}\hat{A}_k. \tag{19}$$

Combining with (19), we get the following estimate

$$\sum_{k=1}^{K-1} -\hat{B}_k \le L\eta \sum_{k=1}^{K-1} \hat{A}_k + \frac{2f(\boldsymbol{x}^1)}{\eta}. \tag{20}$$

On the other hand, with Lemma 3, we have the following bound

$$\frac{2}{1-\theta} \sum_{k=1}^{K} \hat{C}_k \le \frac{2\theta\eta}{1-\theta} \sum_{k=1}^{K} \hat{A}_k + \frac{6B^2}{\sqrt{\delta}} + (2L^2/\sqrt{\delta} + L^2/\delta + 5/2)\mathbb{E}\ln(\frac{v^K}{\delta}). \tag{21}$$

Using [Lemma 2, (Li & Orabona, 2019)] and Lemma 6, we have

$$\sum_{k=1}^{K} \hat{A}_k \le \mathbb{E}\ln(\frac{v^K}{\delta}). \tag{22}$$

Substituting (22), (21) and (20) into (18), then we get

$$\sum_{k=1}^{K} \mathbb{E}\Big(\|\nabla f(\boldsymbol{x}^k)\|^2/(v^k)^{\frac{1}{2}}\Big)$$
$$\le \Big(\frac{(1+\theta)\eta}{1-\theta} + 2L^2/\sqrt{\delta} + L^2/\delta + 5/2\Big)\mathbb{E}\ln(\frac{v^K}{\delta}) + \frac{(7-6\theta)B^2}{(1-\theta)\sqrt{\delta}} + \frac{2f(\boldsymbol{x}^1)}{\eta}. \tag{23}$$

Notice that $\frac{1}{(v^k)^{\frac{1}{2}}} \ge \frac{1}{Ck^\alpha}$, then we have

$$\sum_{k=1}^{K} \mathbb{E}\Big(\|[\nabla f(\boldsymbol{x}^k)]^2/(v^k)^{\frac{1}{2}}\|_1\Big) \ge (\sum_{k=1}^{K} \frac{1}{k^\alpha C}) \cdot \min_{1 \le k \le K}\{\mathbb{E}\|\nabla f(\boldsymbol{x}^k)\|^2\}.$$

We then complete the proof by using the fact that $0 < \frac{1}{1-\alpha} \le 2$ when $0 < \alpha \le 1/2$. Thus, we can get the following estimate

$$\min_{1 \le k \le K}\{\mathbb{E}\|\nabla f(\boldsymbol{x}^k)\|^2\} \le \frac{c_3 + c_4(K)}{K^{1-\alpha}},$$

where

$$c_3 := \frac{2(7-6\theta)B^2C}{(1-\theta)\sqrt{\delta}} + \frac{4Cf(\boldsymbol{x}^1)}{\eta},$$

and

$$c_4(K) := \Big(\frac{(1+\theta)\eta}{1-\theta} + 2L^2/\sqrt{\delta} + L^2/\delta + 5/2\Big)\ln(\frac{CK}{\delta}).$$

## D   Proofs of the Technical Lemmas

### D.1   Proof of Lemma 2

With the fact that $\boldsymbol{m}^k = (1-\theta)\sum_{j=1}^{k} \theta^{k-j}\boldsymbol{g}^j$ when $k \ge 1$, we have

$$\|[\boldsymbol{m}^k]^2/(v^k)^{\frac{1}{2}}\| = \sum_{i=1}^{d} |\boldsymbol{m}_i^k/(v^k)^{\frac{1}{4}}|^2 \le \sum_{i=1}^{d}(1-\theta)^2|\sum_{j=1}^{k} \theta^{k-j}\boldsymbol{g}_i^j/(v^k)^{\frac{1}{4}}|^2$$

$$\overset{a)}{\le} \sum_{i=1}^{d}(1-\theta)^2(\sum_{j=1}^{k} \theta^{k-j}(v^k)^{\frac{1}{2}}) \times \sum_{j=1}^{k} \theta^{k-j}(\boldsymbol{g}_i^j)^2/(v^k)$$

$$\le \sum_{i=1}^{d}(1-\theta)^2 \cdot \frac{(v^k)^{\frac{1}{2}}}{1-\theta} \cdot \sum_{j=1}^{k} \theta^{k-j}(\boldsymbol{g}_i^j)^2/(v^k)$$

$$= (1-\theta) \cdot \sum_{j=1}^{k} \theta^{k-j}\|\boldsymbol{g}^j\|^2/(v^k)^{\frac{1}{2}} \overset{b)}{\le} (1-\theta) \cdot \sum_{j=1}^{k} \theta^{k-j}\|\boldsymbol{g}^j\|^2/(v^j)^{\frac{1}{2}}$$

where $a$) uses the Cauchy's inequality $(\sum_{j=1}^{k} a_j b_j)^2 \leq (\sum_{j=1}^{k} a_j^2) \cdot (\sum_{j=1}^{k} b_j^2)$ with $a_j = \theta^{\frac{k-j}{2}}(v^k)^{\frac{1}{4}}$ and $b_j = \theta^{\frac{k-j}{2}} g_i^j/(v^k)^{\frac{1}{2}}$; $b$) is due to $v^j \leq v^k$ when $j \leq k$. Thus, we are led to

$$\sum_{k=1}^{K}\sum_{j=1}^{k} \theta^{k-j}\|\boldsymbol{g}^j\|^2/(v^j)^{\frac{1}{2}} = \sum_{j=1}^{K}\sum_{k=j}^{K} \theta^{k-j}\|\boldsymbol{g}^j\|^2/(v^j)^{\frac{1}{2}}$$

$$= \sum_{j=1}^{K}\sum_{k=j}^{K} \theta^{k-j}\|\boldsymbol{g}^j\|^2/(v^j)^{\frac{1}{2}} \leq \frac{1}{1-\theta}\sum_{j=1}^{K} \|\boldsymbol{g}^j\|^2/(v^j)^{\frac{1}{2}}.$$

Thus, we can get

$$\sum_{k=1}^{K} A_k \leq \sum_{k=1}^{K} \mathbb{E}[\|\boldsymbol{g}^k\|^2/(v^k)^{\frac{1}{2}}].$$

Using Lemma 1 with $a_k = \|\boldsymbol{g}^k\|^2$ and $h(\cdot) = \frac{1}{\sqrt{\cdot}}$, we get

$$\sum_{k=1}^{K} \|\boldsymbol{g}^k\|^2/(v^k)^{\frac{1}{2}} \leq \sqrt{v^K} - \sqrt{v^0} \leq \sqrt{v^K},$$

where we used the fact that $v^0 \geq 0$. The proof is then completed.

### D.2  Proof of Lemma 3

Similar to the proof of Lemma 2, we have

$$\|\boldsymbol{x}^{k+1} - \boldsymbol{x}^k\|^2 = \|\mathbf{Proj}_{\mathcal{K}}(\boldsymbol{x}^k - \eta\boldsymbol{m}^k) - \boldsymbol{x}^k\|^2 = \|\mathbf{Proj}_{\mathcal{K}}(\boldsymbol{x}^k - \eta\boldsymbol{m}^k) - \mathbf{Proj}_{\mathcal{K}}(\boldsymbol{x}^k)\|^2$$

$$\leq \|\boldsymbol{m}^k/(v^k)^{\frac{1}{2}}\|^2 = \sum_{i=1}^{d} |m_i^k/(v^k)^{\frac{1}{2}}|^2 \leq \sum_{i=1}^{d}(1-\theta)^2|\sum_{j=1}^{k}\theta^{k-j}g_i^j/(v^k)^{\frac{1}{2}}|^2$$

$$\overset{a)}{\leq} \sum_{i=1}^{d}(1-\theta)^2(\sum_{j=1}^{k}\theta^{k-j}) \cdot \sum_{j=1}^{k}\theta^{k-j}\frac{(g_i^j)^2}{(v^k)} \leq \sum_{i=1}^{d}(1-\theta)^2 \cdot \frac{1}{1-\theta} \cdot \sum_{j=1}^{k-1}\theta^{k-j}(g_i^j)^2/(v^k)$$

$$= (1-\theta) \cdot \sum_{j=1}^{k}\theta^{k-j}\|\boldsymbol{g}^j\|^2/(v^k) \overset{b)}{=} (1-\theta) \cdot \sum_{j=1}^{k}\theta^{k-j}\|\boldsymbol{g}^j\|^2/v^j$$

where $a$) uses the fact that $(\sum_{j=1}^{k} a_j b_j)^2 \leq \sum_{j=1}^{k} a_j^2 \sum_{j=1}^{k} b_j^2$ with $a_j = \theta^{\frac{k-j}{2}}$ and $b_j = \theta^{\frac{k-j}{2}} g_i^j/(v^k)^{\frac{1}{2}}$, and $b$) is due to $v^j \leq v^k$ when $j \leq k$. Thus, we can get

$$\sum_{k=1}^{K} \|\boldsymbol{x}^k - \boldsymbol{x}^{k-1}\|^2 \leq \sum_{k=1}^{K} \|\boldsymbol{g}^k\|^2/v^k.$$

Using Lemma 1 with $a_k = \|\boldsymbol{g}^k\|^2$ and $h(\cdot) = \frac{1}{\cdot}$, we are then led to

$$\sum_{k=1}^{K} \|\boldsymbol{g}^k\|^2/v^k \leq \ln\frac{v^k}{v^0} \leq \ln\frac{v^k}{\delta}.$$

The proof is then completed.

### D.3 Proof of Lemma 4

Direct computations give us

$$
\begin{aligned}
\mathbb{E}\|\boldsymbol{g}^k\|^2 &= \mathbb{E}\|\nabla F(\boldsymbol{x}^k;\xi^k,\xi^{k-1})\|^2 \\
&= \mathbb{E}\|\nabla F(\boldsymbol{x}^k;\xi^k,\xi^{k-1}) - \nabla F(\boldsymbol{x}^{k-1};\xi^k,\xi^{k-1}) + \nabla F(\boldsymbol{x}^{k-1};\xi^k,\xi^{k-1})\|^2 \\
&= \mathbb{E}\|\nabla F(\boldsymbol{x}^{k-1};\xi^k,\xi^{k-1})\|^2 + \mathbb{E}\|\nabla F(\boldsymbol{x}^k;\xi^k,\xi^{k-1}) - \nabla F(\boldsymbol{x}^{k-1};\xi^k,\xi^{k-1})\|^2 \\
&\quad + 2\mathbb{E}\langle \nabla F(\boldsymbol{x}^k;\xi^k,\xi^{k-1}) - \nabla F(\boldsymbol{x}^{k-1};\xi^k,\xi^{k-1}), \nabla F(\boldsymbol{x}^{k-1};\xi^k,\xi^{k-1})\rangle \\
&\stackrel{a)}{\geq} \frac{1}{2}\mathbb{E}\|\nabla F(\boldsymbol{x}^{k-1};\xi^k,\xi^{k-1})\|^2 - \mathbb{E}\|\nabla F(\boldsymbol{x}^k;\xi^k,\xi^{k-1}) - \nabla F(\boldsymbol{x}^{k-1};\xi^k,\xi^{k-1})\|^2 \\
&\geq \frac{1}{2}\mathbb{E}\|\nabla F(\boldsymbol{x}^{k-1};\xi^k,\xi^{k-1})\|^2 - L^2\mathbb{E}\|\boldsymbol{x}^k - \boldsymbol{x}^{k-1}\|^2,
\end{aligned}
$$

where $a)$ uses the inequality $2\mathbb{E}\langle \nabla F(\boldsymbol{x}^k;\xi^k,\xi^{k-1}) - \nabla F(\boldsymbol{x}^{k-1};\xi^k,\xi^{k-1}), \nabla F(\boldsymbol{x}^{k-1};\xi^k,\xi^{k-1})\rangle \geq -\frac{1}{2}\mathbb{E}\|\nabla F(\boldsymbol{x}^{k-1};\xi^k,\xi^{k-1})\|^2 - 2\mathbb{E}\|\nabla F(\boldsymbol{x}^k;\xi^k,\xi^{k-1}) - \nabla F(\boldsymbol{x}^{k-1};\xi^k,\xi^{k-1})\|^2$. Thus we can get

$$
\mathbb{E}\|\nabla F(\boldsymbol{x}^{k-1};\xi^k,\xi^{k-1})\|^2 \leq 2L^2\mathbb{E}\|\boldsymbol{x}^k - \boldsymbol{x}^{k-1}\|^2 + 2\mathbb{E}\|\boldsymbol{g}^k\|^2.
$$

### D.4 Proof of Lemma 5

The convexity of $f_{i_k}(\boldsymbol{x})$ with respect to $\boldsymbol{x}$ and $\boldsymbol{g}^k = \nabla F(\boldsymbol{x}^k;\xi^k,\xi^{k-1})$ gives us

$$
\begin{aligned}
\mathbb{E}\langle \boldsymbol{x}^* - \boldsymbol{x}^k, \boldsymbol{g}^k\rangle &\leq \mathbb{E}[F(\boldsymbol{x}^*;\xi^k,\xi^{k-1}) - F(\boldsymbol{x}^k;\xi^k,\xi^{k-1})] \\
&= \mathbb{E}[F(\boldsymbol{x}^{k-1};\xi^k,\xi^{k-1}) - F(\boldsymbol{x}^k;\xi^k,\xi^{k-1}) + F(\boldsymbol{x}^*;\xi^k,\xi^{k-1}) - F(\boldsymbol{x}^{k-1};\xi^k,\xi^{k-1})]. \quad (24)
\end{aligned}
$$

The Lipchitz gradient continuity of $F(\boldsymbol{x};\xi^k,\xi^{k-1})$ with respect to $\boldsymbol{x}$ indicates

$$
\begin{aligned}
&\mathbb{E}[F(\boldsymbol{x}^{k-1};\xi^k,\xi^{k-1}) - F(\boldsymbol{x}^k;\xi^k,\xi^{k-1})] \\
&\leq \frac{L}{2}\mathbb{E}\|\boldsymbol{x}^k - \boldsymbol{x}^{k-1}\|^2 + \mathbb{E}\langle \boldsymbol{x}^{k-1} - \boldsymbol{x}^k, \nabla F(\boldsymbol{x}^k;\xi^k,\xi^{k-1})\rangle \\
&\leq \frac{L}{2}\mathbb{E}\|\boldsymbol{x}^k - \boldsymbol{x}^{k-1}\|^2 + \mathbb{E}\langle \boldsymbol{x}^{k-1} - \boldsymbol{x}^k, \nabla F(\boldsymbol{x}^{k-1};\xi^k,\xi^{k-1})\rangle \\
&\quad + \mathbb{E}\langle \boldsymbol{x}^{k-1} - \boldsymbol{x}^k, \nabla F(\boldsymbol{x}^k;\xi^k,\xi^{k-1}) - \nabla F(\boldsymbol{x}^{k-1};\xi^k,\xi^{k-1})\rangle \\
&\leq \frac{3L}{2}\mathbb{E}\|\boldsymbol{x}^k - \boldsymbol{x}^{k-1}\|^2 + \mathbb{E}\langle \boldsymbol{x}^{k-1} - \boldsymbol{x}^k, \nabla F(\boldsymbol{x}^{k-1};\xi^k,\xi^{k-1})\rangle,
\end{aligned} \quad (25)
$$

where we used the Cauchy's inequality $\mathbb{E}\langle \boldsymbol{x}^{k-1} - \boldsymbol{x}^k, \nabla F(\boldsymbol{x}^k;\xi^k,\xi^{k-1}) - \nabla F(\boldsymbol{x}^{k-1};\xi^k,\xi^{k-1})\rangle \leq L\|\boldsymbol{x}^{k-1} - \boldsymbol{x}^k\|^2$. Because $\xi^k,\xi^{k-1}$ are independent of $\boldsymbol{x}^{k-1}$,

$$
\mathbb{E}(F(\boldsymbol{x}^*;\xi^k,\xi^{k-1}) - F(\boldsymbol{x}^{k-1};\xi^k,\xi^{k-1})) = f(\boldsymbol{x}^*) - f(\boldsymbol{x}^{k-1}). \quad (26)
$$

Turning back to (24), we get

$$
\mathbb{E}\langle \boldsymbol{x}^* - \boldsymbol{x}^k, \boldsymbol{g}^k\rangle \leq \mathbb{E}[F(\boldsymbol{x}^{k-1};\xi^k,\xi^{k-1}) - F(\boldsymbol{x}^k;\xi^k,\xi^{k-1})] + f(\boldsymbol{x}^*) - f(\boldsymbol{x}^{k-1}). \quad (27)
$$

Notice that $\boldsymbol{x}^{k-1}$ is independent of $(\xi^k,\xi^{k-1})$,

$$
\begin{aligned}
&\mathbb{E}\langle \boldsymbol{x}^{k-1} - \boldsymbol{x}^k, \nabla F(\boldsymbol{x}^{k-1};\xi^k,\xi^{k-1})\rangle \\
&= \mathbb{E}\langle \boldsymbol{x}^{k-1} - \boldsymbol{x}^k, \nabla f(\boldsymbol{x}^{k-1})\rangle \\
&\quad + \mathbb{E}\langle \boldsymbol{x}^{k-1} - \boldsymbol{x}^k, \nabla F(\boldsymbol{x}^{k-1};\xi^k,\xi^{k-1}) - \mathbb{E}\nabla F(\boldsymbol{x}^{k-1};\xi^k,\xi^{k-1})\rangle \\
&\leq \mathbb{E}\langle \boldsymbol{x}^{k-1} - \boldsymbol{x}^k, \nabla f(\boldsymbol{x}^{k-1})\rangle + \frac{1}{2}\mathbb{E}\frac{\|\boldsymbol{m}^{k-1}\|^2}{\sqrt{v^{k-1}}} \\
&\quad + \frac{1}{2}\mathbb{E}\frac{\|\nabla F(\boldsymbol{x}^{k-1};\xi^k,\xi^{k-1}) - \mathbb{E}\nabla F(\boldsymbol{x}^{k-1};\xi^k,\xi^{k-1})\|^2}{\sqrt{v^{k-1}}},
\end{aligned} \quad (28)
$$

where we used $|\mathbb{E}\langle X, Y\rangle| \leq \mathbb{E}\|X\|^2 + \mathbb{E}\|Y\|^2$ with $X = \sqrt[4]{v^{k-1}}(\boldsymbol{x}^{k-1} - \boldsymbol{x}^k)$, and $Y = [\nabla F(\boldsymbol{x}^{k-1}; \xi^k, \xi^{k-1}) - \mathbb{E}\nabla F(\boldsymbol{x}^{k-1}; \xi^k, \xi^{k-1})]/\sqrt[4]{v^{k-1}}$. Now, we turn to the upper bound of $\frac{1}{2}\mathbb{E}\frac{\|\nabla F(\boldsymbol{x}^{k-1}; \xi^k, \xi^{k-1}) - \mathbb{E}\nabla F(\boldsymbol{x}^{k-1}; \xi^k, \xi^{k-1})\|^2}{\sqrt{v^{k-1}}}$:

$$
\begin{aligned}
&\frac{1}{2}\mathbb{E}\frac{\|\nabla F(\boldsymbol{x}^{k-1}; \xi^k, \xi^{k-1}) - \mathbb{E}\nabla F(\boldsymbol{x}^{k-1}; \xi^k, \xi^{k-1})\|^2}{\sqrt{v^{k-1}}} \\
&= \frac{1}{2}\mathbb{E}\frac{\|\nabla F(\boldsymbol{x}^{k-1}; \xi^k, \xi^{k-1}) - \mathbb{E}\nabla F(\boldsymbol{x}^{k-1}; \xi^k, \xi^{k-1})\|^2}{\sqrt{v^{k-2}}} \\
&\quad + \frac{1}{2}\mathbb{E}\left[(\|\nabla F(\boldsymbol{x}^{k-1}; \xi^k, \xi^{k-1}) - \mathbb{E}\nabla F(\boldsymbol{x}^{k-1}; \xi^k, \xi^{k-1})\|^2) \times (\frac{1}{\sqrt{v^{k-1}}} - \frac{1}{\sqrt{v^{k-2}}})\right] \\
&\leq \frac{1}{2}\frac{\mathbb{E}\|\nabla F(\boldsymbol{x}^{k-1}; \xi^k, \xi^{k-1})\|^2}{\sqrt{v^{k-2}}} + 2B^2\mathbb{E}(\frac{1}{\sqrt{v^{k-2}}} - \frac{1}{\sqrt{v^{k-1}}}) \\
&\stackrel{a)}{\leq} \frac{L^2\mathbb{E}\|\boldsymbol{x}^k - \boldsymbol{x}^{k-1}\|^2 + \mathbb{E}\|\boldsymbol{g}^k\|^2}{\sqrt{v^{k-2}}} + 2B^2\mathbb{E}(\frac{1}{\sqrt{v^{k-2}}} - \frac{1}{\sqrt{v^{k-1}}}) \\
&\leq \phi_k,
\end{aligned}
\tag{29}
$$

where $\phi_k := \mathbb{E}\frac{\|\boldsymbol{g}^k\|^2}{\sqrt{v^k}} + 2B^2\mathbb{E}(\frac{1}{\sqrt{v^{k-2}}} - \frac{1}{\sqrt{v^{k-1}}}) + \frac{L^2\mathbb{E}\|\boldsymbol{x}^k - \boldsymbol{x}^{k-1}\|^2}{\sqrt{\delta}} + B^2\mathbb{E}(\frac{1}{\sqrt{v^{k-2}}} - \frac{1}{\sqrt{v^k}})$, and $a)$ depends on Lemma 4. Thus, we have

$$
\begin{aligned}
&\mathbb{E}\langle \boldsymbol{x}^{k-1} - \boldsymbol{x}^k, \nabla F(\boldsymbol{x}^{k-1}; \xi^k, \xi^{k-1})\rangle \\
&\leq \mathbb{E}\langle \boldsymbol{x}^{k-1} - \boldsymbol{x}^k, \nabla f(\boldsymbol{x}^{k-1})\rangle + \frac{1}{2}\mathbb{E}\frac{\|\boldsymbol{m}^{k-1}\|^2}{\sqrt{v^{k-1}}} + \phi_k.
\end{aligned}
\tag{30}
$$

Once with the Lipchitz property,

$$
\langle \nabla f(\boldsymbol{x}^{k-1}), \boldsymbol{x}^{k-1} - \boldsymbol{x}^k\rangle \leq f(\boldsymbol{x}^{k-1}) - f(\boldsymbol{x}^k) + \frac{L}{2}\|\boldsymbol{x}^k - \boldsymbol{x}^{k-1}\|^2.
\tag{31}
$$

Combing (31) and (30), we then get

$$
\begin{aligned}
\mathbb{E}\langle \boldsymbol{x}^{k-1} - \boldsymbol{x}^k, \nabla F(\boldsymbol{x}^{k-1}; \xi^k, \xi^{k-1})\rangle &\leq f(\boldsymbol{x}^{k-1}) - f(\boldsymbol{x}^k) \\
&+ \frac{L}{2}\|\boldsymbol{x}^k - \boldsymbol{x}^{k-1}\|^2 + \frac{1}{2}\mathbb{E}\frac{\|\boldsymbol{m}^{k-1}\|^2}{\sqrt{v^{k-1}}} + \phi_k.
\end{aligned}
\tag{32}
$$

Substituting (32) into (25),

$$
\begin{aligned}
\mathbb{E}[F(\boldsymbol{x}^{k-1}; \xi^k, \xi^{k-1}) - F(\boldsymbol{x}^k; \xi^k, \xi^{k-1})] &\leq 2L\mathbb{E}\|\boldsymbol{x}^k - \boldsymbol{x}^{k-1}\|^2 \\
&+ f(\boldsymbol{x}^{k-1}) - f(\boldsymbol{x}^k) + \frac{1}{2}\mathbb{E}\frac{\|\boldsymbol{m}^{k-1}\|^2}{\sqrt{v^{k-1}}} + \phi_k.
\end{aligned}
\tag{33}
$$

Substituting (33) into (27), we then get

$$
\mathbb{E}\langle \boldsymbol{x}^* - \boldsymbol{x}^k, \boldsymbol{g}^k\rangle \leq 2L\mathbb{E}\|\boldsymbol{x}^k - \boldsymbol{x}^{k-1}\|^2 + f(\boldsymbol{x}^*) - f(\boldsymbol{x}^k) + \frac{1}{2}\mathbb{E}\frac{\|\boldsymbol{m}^{k-1}\|^2}{\sqrt{v^{k-1}}} + \phi_k.
\tag{34}
$$

According to our algorithm and we denote $\Lambda := \mathbb{E}(\langle \boldsymbol{x}^* - \boldsymbol{x}^k, \boldsymbol{g}^k\rangle)$, then we have

$$
\begin{aligned}
\mathbb{E}\left(\langle \boldsymbol{x}^* - \boldsymbol{x}^k, \boldsymbol{m}^k\rangle\right) &= \mathbb{E}\left(\langle \boldsymbol{x}^* - \boldsymbol{x}^k, \theta\boldsymbol{m}^{k-1} + (1-\theta)\boldsymbol{g}^k\rangle\right) \\
&= (1-\theta)\cdot\Lambda + \theta\mathbb{E}\langle \boldsymbol{x}^* - \boldsymbol{x}^k, \boldsymbol{m}^{k-1}\rangle \\
&= (1-\theta)\cdot\Lambda + \theta\mathbb{E}\langle \boldsymbol{x}^* - \boldsymbol{x}^{k-1}, \boldsymbol{m}^{k-1}\rangle + \theta\mathbb{E}\langle \boldsymbol{x}^k - \boldsymbol{x}^{k-1}, \boldsymbol{m}^{k-1}\rangle \\
&\stackrel{b)}{\leq} (1-\theta)\cdot\Lambda + \theta\mathbb{E}\langle \boldsymbol{x}^* - \boldsymbol{x}^{k-1}, \boldsymbol{m}^{k-1}\rangle + \eta\theta\mathbb{E}\|\boldsymbol{m}^{k-1}\|^2/(v^{k-1})^{\frac{1}{2}},
\end{aligned}
$$

where $b)$ depends on that $\langle \boldsymbol{x}^k - \boldsymbol{x}^{k-1}, \boldsymbol{m}^{k-1}\rangle \leq \|\boldsymbol{x}^k - \boldsymbol{x}^{k-1}\|\cdot\|\boldsymbol{m}^{k-1}\| = \|\mathbf{Proj}_{\mathcal{K}}(\boldsymbol{x}^{k-1} - \eta\boldsymbol{m}^k) - \mathbf{Proj}_{\mathcal{K}}(\boldsymbol{x}^{k-1})\|\cdot\|\boldsymbol{m}^{k-1}\| \leq \|\boldsymbol{m}^{k-1}\|^2/(v^{k-1})^{\frac{1}{2}}$. Then, we get

$$
B_k \leq (1-\theta)\mathbb{E}\langle \boldsymbol{x}^* - \boldsymbol{x}^k, \boldsymbol{g}^k\rangle + \theta B_{k-1} + \eta\theta A_{k-1}.
\tag{35}
$$

Substituting (34) into (35), we then proved the desired result.

### D.5 Proof of Lemma 6

This proof is identical to the proof of Lemma 3 and will not be reproduced.

### D.6 Proof of Lemma 7

Notice that $\mathbb{E}\nabla F(\boldsymbol{x}^{k-1}; \xi^k, \xi^{k-1}) = \nabla f(\boldsymbol{x}^{k-1})$, we then get

$$
\begin{aligned}
&\mathbb{E}\langle -\nabla f(\boldsymbol{x}^k)/(v^k)^{1/2}, \boldsymbol{g}^k\rangle \\
&= \mathbb{E}\langle -\nabla f(\boldsymbol{x}^k)/(v^k)^{1/2}, \nabla F(\boldsymbol{x}^k; \xi^k, \xi^{k-1})\rangle \\
&= -\mathbb{E}\|\nabla f(\boldsymbol{x}^k)\|^2/(v^k)^{1/2} + \mathbb{E}\langle \nabla f(\boldsymbol{x}^k)/(v^k)^{\frac{1}{2}}, \nabla f(\boldsymbol{x}^k) - \nabla f(\boldsymbol{x}^{k-1})\rangle \\
&\quad + \underbrace{\mathbb{E}\langle \nabla f(\boldsymbol{x}^k)/(v^k)^{\frac{1}{2}}, \nabla f(\boldsymbol{x}^{k-1}) - \nabla F(\boldsymbol{x}^{k-1}; \xi^k, \xi^{k-1})\rangle}_{:=(\dagger)} \\
&\quad + \mathbb{E}\langle \frac{\nabla f(\boldsymbol{x}^k)}{(v^k)^{\frac{1}{2}}}, \nabla F(\boldsymbol{x}^{k-1}; \xi^k, \xi^{k-1}) - \nabla F(\boldsymbol{x}^k; \xi^k, \xi^{k-1})\rangle.
\end{aligned}
\tag{36}
$$

The Cauchy's inequality together with the smooth assumption gives us

$$
\begin{aligned}
&\mathbb{E}\langle \nabla f(\boldsymbol{x}^k)/(v^k)^{\frac{1}{2}}, \nabla f(\boldsymbol{x}^k) - \nabla f(\boldsymbol{x}^{k-1})\rangle \\
&\leq |\mathbb{E}\langle \nabla f(\boldsymbol{x}^k)/(v^k)^{\frac{1}{4}}, [\nabla f(\boldsymbol{x}^{k-1}) - \nabla f(\boldsymbol{x}^k)]/(v^k)^{\frac{1}{4}}\rangle| \\
&\leq \frac{1}{4}\mathbb{E}\|\nabla f(\boldsymbol{x}^k)\|^2/(v^k)^{\frac{1}{2}} + \mathbb{E}\|\nabla f(\boldsymbol{x}^{k-1}) - \nabla f(\boldsymbol{x}^k)\|^2/(v^k)^{\frac{1}{2}} \\
&\leq \frac{1}{4}\mathbb{E}\|\nabla f(\boldsymbol{x}^k)\|^2/(v^k)^{\frac{1}{2}} + L^2\mathbb{E}\|\boldsymbol{x}^{k-1} - \boldsymbol{x}^k\|^2/(v^k)^{\frac{1}{2}}.
\end{aligned}
\tag{37}
$$

Similarly, the Lipschitz property of the stochastic gradient yields

$$
\begin{aligned}
&\mathbb{E}\langle \nabla f(\boldsymbol{x}^k)/(v^k)^{\frac{1}{2}}, \nabla F(\boldsymbol{x}^{k-1}; \xi^k, \xi^{k-1}) - \nabla F(\boldsymbol{x}^k; \xi^k, \xi^{k-1})\rangle \\
&\leq \mathbb{E}|\langle \frac{\nabla f(\boldsymbol{x}^k)}{(v^k)^{\frac{1}{4}}}, \frac{[\nabla F(\boldsymbol{x}^{k-1}; \xi^k, \xi^{k-1}) - \nabla F(\boldsymbol{x}^k; \xi^k, \xi^{k-1})]}{(v^k)^{\frac{1}{4}}}\rangle| \\
&\leq \frac{1}{4}\mathbb{E}\|\nabla f(\boldsymbol{x}^k)\|^2/(v^k)^{\frac{1}{2}} + L^2\mathbb{E}\|\boldsymbol{x}^{k-1} - \boldsymbol{x}^k\|^2/(v^k)^{\frac{1}{2}}.
\end{aligned}
\tag{38}
$$

Substituting (38) and (37) into (36), we are then led to

$$
\begin{aligned}
\mathbb{E}\langle -\nabla f(\boldsymbol{x}^k)/(v^k)^{\frac{1}{2}}, \boldsymbol{g}^k\rangle &\leq -\mathbb{E}\Big(\|\nabla f(\boldsymbol{x}^k)\|^2/(v^k)^{\frac{1}{2}}\Big) + (\dagger) \\
&\quad + \frac{1}{2}\mathbb{E}(\|\nabla f(\boldsymbol{x}^k)\|^2/(v^k)^{\frac{1}{2}}) + 2L^2\mathbb{E}(\|\boldsymbol{x}^{k-1} - \boldsymbol{x}^k\|^2/(v^k)^{\frac{1}{2}}).
\end{aligned}
\tag{39}
$$

Now, we turn to bound $(\dagger)$:

$$
\begin{aligned}
(\dagger) &= \underbrace{\mathbb{E}\langle \nabla f(\boldsymbol{x}^{k-1})/(v^{k-1})^{\frac{1}{2}}, \nabla f(\boldsymbol{x}^{k-1}) - \nabla F(\boldsymbol{x}^{k-1}; \xi^k, \xi^{k-1})\rangle}_{=0} \\
&\quad + \mathbb{E}\Big\langle \nabla f(\boldsymbol{x}^k)/(v^k)^{\frac{1}{2}} - \nabla f(\boldsymbol{x}^{k-1})/(v^k)^{\frac{1}{2}} \\
&\qquad + \nabla f(\boldsymbol{x}^{k-1})/(v^k)^{\frac{1}{2}} - \nabla f(\boldsymbol{x}^{k-1})/(v^{k-1})^{\frac{1}{2}}, \nabla f(\boldsymbol{x}^{k-1}) - \nabla F(\boldsymbol{x}^{k-1}; \xi^k, \xi^{k-1})\Big\rangle \\
&\leq \mathbb{E}\langle [\nabla f(\boldsymbol{x}^k) - \nabla f(\boldsymbol{x}^{k-1})]/(v^k)^{\frac{1}{2}}, \nabla f(\boldsymbol{x}^{k-1}) - \nabla F(\boldsymbol{x}^{k-1}; \xi^k, \xi^{k-1})\rangle \\
&\quad + 2B^2\mathbb{E}[1/(v^{k-2})^{\frac{1}{2}} - 1/(v^k)^{\frac{1}{2}}] \\
&\leq \frac{\mathbb{E}\|\boldsymbol{x}^k - \boldsymbol{x}^{k-1}\|^2}{2} + 2B^2\mathbb{E}[1/(v^{k-2})^{\frac{1}{2}} - 1/(v^k)^{\frac{1}{2}}] \\
&\quad + \frac{1}{2}\mathbb{E}\|\nabla f(\boldsymbol{x}^{k-1}) - \nabla F(\boldsymbol{x}^{k-1}; \xi^k, \xi^{k-1})\|^2/v^k.
\end{aligned}
$$

Furthermore, we have

$$
\frac{1}{2}\mathbb{E}\|\nabla f(\boldsymbol{x}^{k-1}) - \nabla F(\boldsymbol{x}^{k-1};\xi^k,\xi^{k-1})\|^2/v^k
$$
$$
= \frac{1}{2}\mathbb{E}\|\nabla f(\boldsymbol{x}^{k-1}) - \nabla F(\boldsymbol{x}^{k-1};\xi^k,\xi^{k-1})\|^2/v^{k-2}
$$
$$
+ \frac{1}{2}\mathbb{E}\|\nabla f(\boldsymbol{x}^{k-1}) - \nabla F(\boldsymbol{x}^{k-1};\xi^k,\xi^{k-1})\|^2(1/v^k - 1/v^{k-2})
$$
$$
\overset{\text{Lemma 4}}{\leq} L^2\mathbb{E}\|\boldsymbol{x}^k - \boldsymbol{x}^{k-1}\|^2/v^{k-2} + 2\mathbb{E}\|\boldsymbol{g}^k\|^2/v^{k-2} + 2B^2\mathbb{E}(1/v^k - 1/v^{k-2})
$$
$$
\leq \frac{L^2}{\delta}\mathbb{E}\|\boldsymbol{x}^k - \boldsymbol{x}^{k-1}\|^2 + 2\mathbb{E}\|\boldsymbol{g}^k\|^2/v^k + 4B^2\mathbb{E}(1/v^k - 1/v^{k-2}).
$$

Thus, (39) can also be bounded as

$$
\begin{aligned}
\mathbb{E}\langle -\nabla f(\boldsymbol{x}^k)/(v^k)^{\frac{1}{2}}, \boldsymbol{g}^k\rangle &\leq -\frac{1}{2}\mathbb{E}\Big(\|\nabla f(\boldsymbol{x}^k)\|^2/(v^k)^{\frac{1}{2}}\Big)\\
&+ 6B^2\mathbb{E}[1/(v^{k-2})^{\frac{1}{2}} - 1/(v^k)^{\frac{1}{2}}] + 2\mathbb{E}\|\boldsymbol{g}^k\|^2/v^k\\
&+ (2L^2/\sqrt{\delta} + L^2/\delta + 1/2)\mathbb{E}\|\boldsymbol{x}^k - \boldsymbol{x}^{k-1}\|^2.
\end{aligned}
\tag{40}
$$

We also use a shorthand notation $\Lambda := \mathbb{E}(\langle -\nabla f(\boldsymbol{x}^k)/(v^k)^{\frac{1}{2}}, \boldsymbol{g}^k\rangle)$ and then

$$
\begin{aligned}
&\mathbb{E}\left(\langle -\nabla f(\boldsymbol{x}^k), \boldsymbol{m}^k/(v^k)^{\frac{1}{2}}\rangle\right)\\
&= \mathbb{E}\left(\langle -\nabla f(\boldsymbol{x}^k)/(v^k)^{\frac{1}{2}}, \theta\boldsymbol{m}^{k-1} + (1-\theta)\boldsymbol{g}^k\rangle\right)\\
&= (1-\theta)\cdot\Lambda + \theta\mathbb{E}\langle -\nabla f(\boldsymbol{x}^k)/(v^k)^{\frac{1}{2}}, \boldsymbol{m}^{k-1}\rangle\\
&\overset{a)}{\leq} (1-\theta)\cdot\Lambda + \theta\mathbb{E}\langle -\nabla f(\boldsymbol{x}^{k-1})/(v^{k-1})^{\frac{1}{2}}, \boldsymbol{m}^{k-1}\rangle + \theta\eta\mathbb{E}\|\boldsymbol{m}^{k-1}\|^2/v^{k-1}.
\end{aligned}
\tag{41}
$$

where $a)$ uses the Cauchy-Schwarz inequality and the Lipschitz property of $f$ and $v^{k-1} \leq v^k$. Substituting (41) into (40), we then proved the result.

# E   Ablation study of the parameter $\theta$

In this section, we investigate the influence of the parameter $\theta$ on the convergence rate of AOGD. The experimental setup adheres to the one detailed in Section 3, focusing on the hinge loss applied to the `ijcnn1` dataset.

We fix the parameters $\eta$ and $R$ at values 1.0 and 10.0, respectively, while varying $\theta$ over the set $\{0.1, 0.3, 0.5, 0.7, 0.9\}$. The rationale behind these chosen values for $\eta$ and $R$ stems from optimization results observed at $\theta = 0.9$, aligning with the configurations presented in Table 2 and Fig. 2.

As depicted in Figure 4, the convergence rate of AOGD across different $\theta$ values is displayed. Notably, the convergence rate appears robust to the specific selection of $\theta$ with slightly better performance observed at an intermediate $\theta$ value, specifically $\theta = 0.7$.

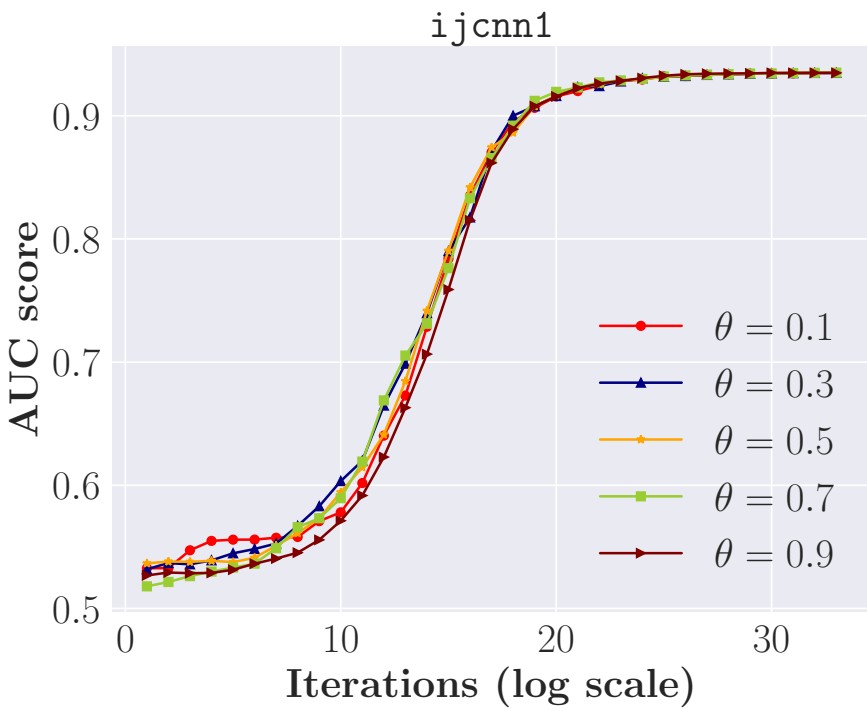

Figure 4: The effect of $\theta$ on the convergence rate of AOGD with hinge loss on `ijcnn1` dataset. The result suggests AOGD achieves robust performance across different $\theta$ values.

