# OpenReview forum: "Pairwise Learning with Adaptive Online Gradient Descent"
_TMLR — Accepted by TMLR_

### Review · Reviewer_s7Zb · 2023-07-24

**Summary Of Contributions:**

The paper considers the pairwise learning problem and develops a new Adam-like method based on Online Gradient Descent (Yang et al. (2021b)). The authors support the new method with the theoretical convergence rates in the convex and non-convex settings.

**Audience:**

Yes

**Broader Impact Concerns:**

-

**Claims And Evidence:**

Yes

**Requested Changes:**

-

**Strengths And Weaknesses:**

Strengths:

The authors prove convergence rates and explain regimes when the new method AOGD improves the previous method OGD. I didn't check the proofs, but the theoretical bounds sound to me.

Weaknesses:

My main concern (or question) is why we can't modify Algorithms 1 and 2 and add the following step between the step 1 and 2:

step 1.5: drop $g^k$ from step 1. Receive $\widehat{\xi}^k$ and calculate $g^k = \nabla F(x^k; \widehat{\xi}^k, \xi^{k}).$

Now $g^k$ is unbiased. The idea is to drop 1st, 3rd, 5th, and so on gradients. This way, we can reduce this problem to the classical stochastic gradient problem. Why does this strategy is overlooked in the pairwise learning community?
Yes, the computation complexity will increase by two. However, the theory will be simpler and nicer. Due to the fact that $g^k$ are biased in Algorithms 1 and 2, the theory requires much stronger assumptions than in the classical unbiased case.

---

### Review · Reviewer_aBKL · 2023-08-04

**Summary Of Contributions:**

This paper proposes an adaptive online gradient descent method for pairwise learning. The authors also provide convergence analysis for the proposed method in strongly convex, convex and nonconvex cases. The proposed method outperforms the non-adaptive counterpart on the online AUC maximization experiments.

**Audience:**

Yes

**Broader Impact Concerns:**

no broad impact concerns

**Claims And Evidence:**

Yes

**Requested Changes:**

I suggest authors study how parameter $\theta$ affects the convergence property. For example, the authors may empirically compare the performance of different choices of $\theta$. In addition, the experiments should compare the proposed methods with Ding te al.(2015) since their method is also suitable for AUC maximization.

**Strengths And Weaknesses:**

The paper is mostly clear and well written. The authors overcome the challenges in the analysis and provide convergence rate in strongly convex, convex and nonconvex cases. However, I have following concerns:

1. My first concern is that the convergence analysis does not reflect the advantage of using momentum. In theorem 1 and theorem 3, choosing $\theta=0$ leads to the best convergence rate (c1, c2, c3, c4 all achieve minimum when $\theta=0$). Also, theorem 2 only considers the momentum-free case.

2. The description of assumption 1 is not accurate. It should be assumed that the gradient of F is Lipschitz continuous, not that the function F is Lipschitz continuous.

3. In section 2.3, the paper claims that AOGD achieves the optimal convergence rate of SGD in the strongly convex case. However, the optimal convergence rate of SGD is O(1/K) (Rakhlin et al., 2012) rather than O(log(K)/K). Thus the convergence rate of AOGD is suboptimal.

4. In the empirical comparison, the performance of AOGD and OGD are very close. Also, I think the method proposed by Ding te al.(2015) should be considered as a baseline for the experiments.

---

### Review · Reviewer_DVMN · 2023-08-07

**Summary Of Contributions:**

This paper proposes a new adaptive online gradient descent algorithm for pairwise learning. Specifically, the algorithm leverages momentum and adaptive stepsizes in a similar way as ADAM. The authors provide convergence bounds under strongly convex, convex, and nonconvex cases and test the algorithm on an  AUC maximization task.

**Audience:**

Yes

**Claims And Evidence:**

No

**Requested Changes:**

-	Clearly compare the convergence bounds with previous bounds. Explain how the added components (momentum and adaptive stepsizes) improve the bounds.

-	Add empirical results to verify theorems.

-	Add empirical results on other pairwise learning tasks beyond AUC maximization.

-	Add the related baseline Ding et al. (2015).


**Strengths And Weaknesses:**

Strengths

-	The paper provides comprehensive theoretical results under both convex and nonconvex cases.

-	The proposed algorithm is easy to use.

Weaknesses

-	The theoretical benefit of using momentum theta is unclear. Theorem 2 assumes no momentum (i.e. theta=0) and the bounds in Theorem 1&3 seem to be optimal when theta = 0. Can the authors explain how using momentum improves the convergence bounds?

-	How do the bounds in the paper compare with previous online GD bounds?

-	The projection operator Proj(·) is not defined in the paper.

-	The authors claim that one of the differences compared with Ding et al. (2015) is that they consider general pairwise learning rather than only AUC maximization, yet the experiments only AUC maximization task. It will be better to show empirical results on other pairwise learning tasks to support the claim that the results in the paper apply to general pairwise learning.

-	Since the experiment only includes one AUC maximization task, it is unclear what case (strongly convex/convex/nonconvex) it falls in. I would suggest the authors conduct experiments for each case and empirically verify their three theorems. The current empirical results do not give many insights. It basically shows that AOGD sometime gives a bit of improvement compared to OGD.

-	The baseline in the experiment should include Ding et al. (2015), which is also an adaptive GD algorithm.

-	The authors mention when alpha is less than 0.5, AOGD converges faster than OGD. It is unclear if the experiment belongs to this case and whether we should expect AOGD to converge faster.

---

### Decision · Action_Editor_bt5V · 2023-10-20

**Recommendation:** Accept with minor revision

**Comment:**

Some concerns were raised by the reviewers but these were appropriately addressed by the revision and the author discussion (in particular, improving the experimental section). Two reviewers were leaning accept while one reviewer was leaning reject due to their concerns about the momentum analysis (see next paragraph) and the experiment section (see my comments about it in the "claims and evidence section above").

One remaining concern from the reviewers after the revision is that the theoretical analysis does not highlight an improvement thanks to momentum (the bound is best with zero momentum in Theorem 1 and 3, and Theorem 2 did not consider the momentum). Given that this type of analysis is quite challenging and this phenomenon is typical in the convergence analysis of momentum methods literature, I don't think that this is too problematic.

There was also a detailed discussion with reviewer s7Zb about the alternative sampling scheme which considers two independent samples at every iteration rather than re-using an old sample as in the current paper and in the OGD algorithm. The advantage of this alternative sampling scheme is that the theoretical analysis is simpler (as the gradient is unbiased). But
1) As mentioned by the authors in their rebuttal and in the revision, the "re-using" sampling approach allows to have T iterations with T samples rather than T/2 iterations with the "2 independent samples" approach, which could suggest more progress. This is not a theoretical necessity though, as perhaps T/2 "cleaner samples" could converge faster than T "biased samples". But empirically, the 2 independent samples approach was observed to perform worse in [Zhao et al. 2011], as mentioned in the author revision
2) Moreover, this "re-using" sample approach was already deemed better in the literature (such as in [1] on which the submission heavily builds on).
This reviewer recommended acceptance at the end.

I thus recommend acceptance subject to a minor revision. I detailed the required changes below.

### Required changes:

1) Proposition 1 provides an argument why their adaptive step-size could improve over OGD theoretically. This hinges on the assumption that $E[\sqrt{v_k}] = O(K^\alpha)$ with $\alpha < \frac{1}{2}$. While the authors provide a theoretical justification that this holds for
$\alpha = \frac{1}{2}$, they do not provide any explanation (or empirical evidence) which could indicate that it holds for $\alpha < \frac{1}{2}$. To clarify the claim of Proposition 1, the authors should either provide in their revision more evidence that this can hold; or at least clarify in their text that this is a purely hypothetical analysis, which, while being mentioned being also assumed in the previous literature in Remark 1, still would require additional inquiry / evidence to conclude something.

2) The discussion about the different sampling schemes made me realize a crucial point that is not explicitly mentioned in the paper, but I think it should be discussed in the revision. The formulation for pairwise learning as defined in Eq. (1) seems to implicitly assume that $F(x; \xi, \xi')$ has some kind of "symmetry" in $\xi$ and $\xi'$ in the sense that they play similar role (as you use the same distribution $P$ on $\xi$ and $\xi'$). Yet, the standard AUC formulation (such as appearing e.g. in [Zhao et al. 2011] does not have this property (it treats the positive and the negative classes differently, and these also can have different distributions). [1] provides a formulation where $\xi$ and $\xi'$ can have the same distribution (see the 'AUC maximization paragraph in their Section 2.1 e.g.). But this is using indicator functions and then the F they define will be always zero unless $\xi$ has a positive label and $\xi'$ has a negative label. Supposing I was sampling the points independently, this means that only 1/4 of my iterations would yield a non-zero signal. And in the online learning regime where someone could give me a fixed stream of data-points; suppose that they had sorted the labels beforehand, and had put all the negative labels first and then the positive labels. Then by always considering two consecutive data-points as my (\xi, \xi') pairs, only when I have a (+ve label, -ve label) pair would I have a non-zero signal. In the worst-case setting I just mentioned, this would actually never happen as the only transition is from -ve to +ve (the wrong direction), and thus nothing would be learned with ODG or the proposed algorithm by the author!

In their revision, the authors should discuss this last aspect in more details and how this affects (or not) their algorithm. For better motivation and clarity, they should also explicitly provide the F formulation for the AUC application that they consider (this could be in the introduction e.g. when introducing the pairwise learning framework). Given that they assume iid samples for $\xi_k$, my worst-case example is unlikely; but it would be useful to clarify all this. In particular, it is an interesting feature of their approach for AUC (assuming the same formulation as in [1]) that they can never have two non-zero updates in a row (but can have many zero updates in a row). This point is somewhat puzzling, and it also brings some nuance to their claim that the "re-using" sampling approach yield T iterations out of T samples vs. the "2 samples approach" which yield T/2 iterations. As at least one out of two iterations is zero, the re-using approach here actually has less than T/2 iterations! In contrast, if we allow to use two independent samples per iteration, one could use one sample from the positive class and one independent sample from the negative class to always have a non-zero update at each iteration for the other approach. The gain of one approach vs. the other becomes less clear from this perspective, and all this should be clarified in the revision.

Please explain in OpenReview he high level changes you have made in the revision to help the verification.

**Audience:**

Yes, see above.

**Claims And Evidence:**

This paper is basically a follow-up work of [1] which had considered an online gradient algorithm (OGD) for pairwise learning. In the current submission, the authors propose an adaptive step-size version (scalar AdaGrad-style) + momentum of OGD and provide a non-trivial convergence analysis (the challenges coming from the fact that the gradient estimate is biased due to interactions between the current sample and the previous one appearing in the pairwise objective).

While the empirical section is rather limited as mentioned by reviewer DVMN (in particular, it is only testing on the online AUC maximization task while the authors claimed as a contribution over [Ding et al. 2015] that their approach is more general than just AUC maximization), the claims and evidence were judged appropriate by the reviewers after the revision. The authors re-use a similar experimental setup as [1], and only claim further generality than AUC maximization in their theoretical convergence analysis, not in their empirical section. I also think that the main interest in this work is more on the theoretical side rather than on the empirical side at this stage.

[1] Yang et al. Simple Stochastic and Online Gradient Descent Algorithms for Pairwise Learning, NeurIPS 2021

---

> ### Author Response · Authors · 2023-10-27
> **Reply to the AE**
>
> Dear Editor,
>
> Thank you for your kind comments.
>
> Based on your suggestions, we have made three changes to the paper. Below, we provide some more detailed explanations of these changes:
>
> 1. Note that the condition $\alpha=1/2$ holds due to bounded stochastic gradients. The fast decaying condition $\alpha<1/2$ explains why adaptive stochastic optimization algorithms generally outperform non-adaptive schemes (Reddi et al., 2018; Chen et al., 2018; 2019; Liu et al., 2019). However, the superiority of adaptive SGD over non-adaptive approaches is not well-explained beyond this assumption. While the assumption is commonly used due to sparse stochastic gradients, it is not a direct consequence of $\alpha<1/2$. Instead, the assumption is more of a hypothetical analysis.
>
> 2. We have revised the formulation of the loss function in the AUC revision in Sec. 3, based on (Yang et al., 2021). It is worth noting that the formulation is implicitly utilized in their code~\footnote{https://github.com/zhenhuan-yang/simple-pairwise}, which our implementation is based on. The loss is only activated for consecutive sample pairs with opposite labels, such as $({\bf x}, y)$ and $( {\bf x}', y')$, where $y=1$ and $y'=-1$, or $y=-1$ and $y'=1$.
>
> 3. We provide a clarification on the pairwise model and AUC maximization:
>
>      3.1) The worst-case scenario mentioned above is highly unlikely to occur because we assume that the data is independently and identically distributed from the underlying distribution.
>
>     3.2) In the online setting, the labels are unknown in advance. While the work of (Zhao et al. 2011) discusses the online scenario, they employ an ``update buffer'' data pre-processing step to divide the data and subsequently run the algorithm offline. Therefore, we cannot directly apply the formulation described in (Zhao et al. 2011).To address this, it is more flexible to utilize the formulation presented in (Yang et al., 2021). This formulation is better suited for online learning, as it does not require an offline processing step like the one used by (Zhao et al. 2011).
>
>    3.3) Existing results from (Yang et al., 2021) demonstrate that the pairwise model outperforms previous methods, even when a negative sample is encountered along with a positive sample. This is because previous methods often have high gradient complexity at each iteration, while the online complexity is very small in the proposed formulation.
>
>    In summary, the reformulation in our paper allows for more flexibility in the optimization process and has the potential to yield better results in practical scenarios.
>
> Best,
> The authors